# MMKU-Bench: A Multimodal Update Benchmark for Diverse Visual Knowledge

**Baochen Fu** [* 1 2]  **Yuntao Du** [* 1 2 3]  **Cheng Chang** [1]  **Baihao Jin** [1]  **Wenzhi Deng** [1]  **Muhao Xu** [1]  **Hongmei Yan** [4]
**Weiye Song**[✉ 1]  **Yi Wan**[✉ 1]

## Abstract

As real-world knowledge continues to evolve, the parametric knowledge acquired by multimodal models during pretraining becomes increasingly difficult to remain consistent with real-world knowledge. Existing research on multimodal knowledge updating focuses only on learning previously unknown knowledge, while overlooking the need to update knowledge that the model has already mastered but that later changes; moreover, evaluation is limited to the same modality, lacking a systematic analysis of cross-modal consistency. To address these issues, this paper proposes MMKU-Bench, a comprehensive evaluation benchmark for multimodal knowledge updating, which contains over 25k knowledge instances and more than 49k images, covering two scenarios, updated knowledge and unknown knowledge, thereby enabling comparative analysis of learning across different knowledge types. On this benchmark, we evaluate a variety of representative approaches, including supervised fine-tuning (SFT), reinforcement learning from human feedback (RLHF), and knowledge editing (KE). Experimental results show that SFT and RLHF are prone to catastrophic forgetting, while KE better preserve general capabilities but exhibit clear limitations in continual updating. Overall, MMKU-Bench provides a reliable and comprehensive evaluation benchmark for multimodal knowledge updating, advancing progress in this field. The code and dataset are available at https://github.com/baochenfu/MMKU-Bench.

---

[*]Equal contribution  [1]Shandong University, Jinan, China [2]Joint SDU-NTU Centre for Artificial Intelligence Research (C-FAIR), Jinan, China [3]State Key Laboratory for Novel Software Technology, Nanjing, China [4]Jinzhong Group, Jinan, China. Correspondence to: Weiye Song <songweiye@sdu.edu.cn>, Yi Wan <wanyi@sdu.edu.cn>.

*Proceedings of the 43rd International Conference on Machine Learning*, Seoul, South Korea. PMLR 306, 2026. Copyright 2026 by the author(s).

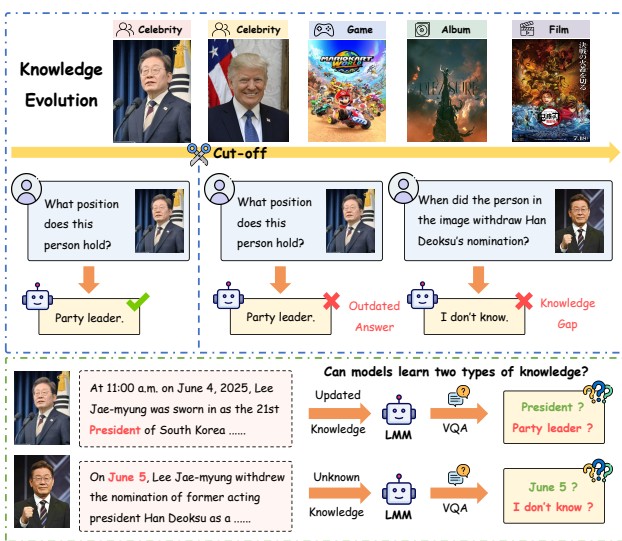

*Figure 1.* After the knowledge cut-off, models suffer from both outdated information and knowledge gaps. Can models learn two types of knowledge?

## 1. Introduction

Large Multimodal Models (LMMs) have achieved remarkable progress in tasks such as visual understanding (Wang et al., 2025), cross-modal reasoning (Yang et al., 2025), and open-ended question answering (Zi et al., 2025). This progress is largely attributed to pretraining on large-scale vision-language data (Bica et al., 2024), which enables models to internalize rich world knowledge within their parameter space and to perform complex reasoning and decision-making accordingly. However, knowledge in the real world is not static (Wang et al., 2024c): facts evolve over time, and information that was once correct can quickly become outdated (Zheng et al., 2023). Under such circumstances, the knowledge encoded in model parameters during pretraining will inevitably drift away from the real world (Khodja et al., 2024), thereby undermining reliability and consistency in practical applications (Ni et al., 2025). This raises a key question: **Can existing methods effectively update the knowledge embedded in LMMs so as to maintain its consistency?**

Recently, several representative works have begun to explore multimodal knowledge injection (Jiang et al., 2025c;

Fu et al., 2025b; Jiang et al.), which primarily focus on the incremental incorporation of newly acquired knowledge after pretraining(Song et al., 2026). In contrast, scenarios in which previously correct knowledge must be updated due to changes in the real world remain underexplored in a systematic manner. From a principled perspective, knowledge updating is not merely a matter of adding new information; rather, it involves rewriting and adjusting existing internal representations (Li et al., 2025; Jiang et al., 2025a), which may affect model behavior in ways fundamentally different from learning previously unknown knowledge. As illustrated in Figure 1, for questions such as "What position does this person hold?", answers that were correct during pretraining may become outdated due to real-world changes; When updated knowledge conflicts with outdated knowledge already stored in the model, it remains unclear whether LMMs can effectively overwrite prior memories and adhere to the updated information. This leads to a second question: **Are there significant differences between updating existing knowledge and injecting previously unknown knowledge in terms of learning difficulty and their impact on model behavior for LMMs?**

Furthermore, regardless of whether the goal is to update existing knowledge or to inject previously unknown knowledge, prior studies(Ding et al., 2022; Lee et al., 2024; Zhang et al., 2025) typically focus on knowledge acquisition and utilization within the same modality. However, in real-world multimodal applications, models often need to acquire or update knowledge in one modality and perform reasoning and decision-making in another. For example, a model may receive updated information in textual form but be required to apply it in visual question answering(Peng et al., 2025). In such cases, whether knowledge can maintain semantically consistent representations and transferable usability across different modalities becomes a critical factor affecting model reliability(Ahmed et al., 2022; Zhu et al., 2025; Liang et al., 2026). Despite its importance, existing research on multimodal knowledge injection and updating lacks a systematic evaluation of cross-modal knowledge transfer. This gap leads to a third core question: **Can large multimodal models achieve stable and consistent cross-modal transfer at the level of knowledge updating?**

To address the above questions, in this paper, we propose MMKU-Bench, a benchmark designed for the systematic evaluation of multimodal knowledge updating. The benchmark not only enables isolated assessment of a model's ability to update knowledge but also supports direct comparison with scenarios involving the injection of previously unknown knowledge while further covering the evaluation of cross-modal knowledge transfer.

To this end, we systematically collect visual question answering (VQA) instances that LMMs can answer entirely

correctly, and build knowledge-updating data upon this foundation, while additionally introducing knowledge that emerged after the model's knowledge cutoff and was previously unknown by the model. Based on these data, we design three complementary evaluation settings(Duan et al., 2024): (1) knowledge injection evaluation, which assesses the ability of existing methods to introduce updated/unknown knowledge into models; (2) general capability evaluation, which measures the preservation of a model's original capabilities and knowledge retention after injection; and (3) consistency evaluation, which evaluates the consistency of model outputs across different modalities and under various perturbations, enabling a comprehensive and systematic comparison of different knowledge types and injection methods.

Extensive experiments show that existing knowledge updating paradigms face several challenges. First, models exhibit pronounced knowledge inertia: semantic priors facilitate learning updated knowledge in QA tasks, whereas in multiple-choice settings, prior knowledge often acts as a strong distractor, leading to poorer performance on updated knowledge than on unknown knowledge. Second, knowledge updating is costly. Although SFT and RLHF can effectively update knowledge, they often degrade general capabilities; compared to unknown knowledge, updating existing knowledge is more likely to disrupt representational stability and induce catastrophic forgetting. Finally, cross-modal analysis shows limited transfer consistency of injected knowledge during cross-modal reasoning, revealing structural deficiencies in current multimodal knowledge alignment mechanisms.

In summary, the main contributions of this paper are as follows:

- We construct MMKU-Bench, a comprehensive benchmark for multimodal knowledge updating, using free-form natural language knowledge to evaluate the effectiveness, robustness, and logical consistency of various knowledge injection methods;

- We conduct extensive experiments on knowledge injection, revealing two major challenges: the poor performance of existing knowledge injection methods and the catastrophic forgetting caused by supervised fine-tuning;

- We provide an in-depth analysis from the perspectives of cross-modal transfer and decision consistency, offering experimental insights and empirical guidance for designing more robust and efficient multimodal knowledge updating mechanisms.

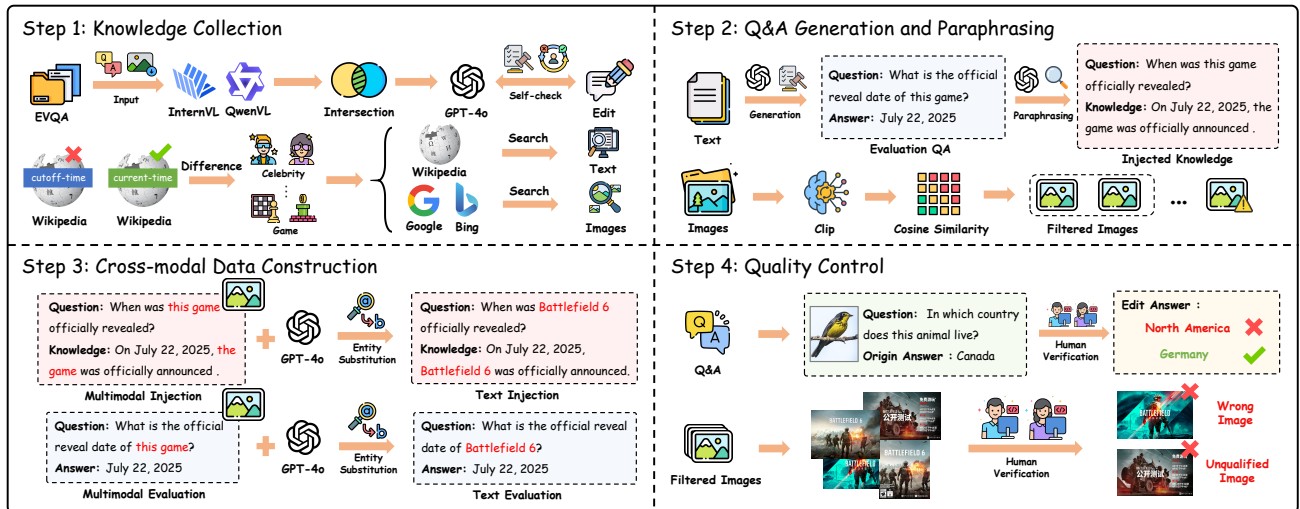

*Figure 2.* Overall pipeline of the construction for MMKU-Bench.

## 2. Related Work

### 2.1. Large Multimodal Model

LMMs integrate vision encoders with large language models to enable cross-modal alignment and understanding. CLIP (Radford et al., 2021) establishes a unified vision–language representation space, while BLIP-2 (Li et al., 2023a) and LLaVA (Liu et al., 2023a) introduce lightweight alignment modules to project visual features into the language embedding space. Recent advances focus on training strategies and visual modeling: LLaVA-OneVision (Li et al., 2024b) employs multi-stage full-parameter training to improve generalization and reasoning, InternVL 2.5 (Chen et al., 2024) enhances robustness in complex scenarios via dynamic high-resolution training, and Qwen2.5 VL (Bai et al., 2025) improves spatial and temporal modeling efficiency through dynamic resolution and MRoPE.

### 2.2. Knowledge Updating Benchmark

As knowledge timeliness becomes increasingly critical, the ability of models to update existing knowledge has attracted growing attention. In the text domain, ZSRE (Yao et al., 2023) and CounterFact (Meng et al., 2022) focus on single-fact editing, while MQuAKE (Zhong et al., 2023) and EvoWiki (Tang et al., 2025) study knowledge updating from multi-hop reasoning and knowledge-type perspectives. In the multimodal setting, VLKEB (Huang et al., 2024) and MMKE-Bench (Du et al., 2025) explore vision–language knowledge editing, and MMEVOKE (Jiang et al.) evaluates models' ability to acquire knowledge beyond the training cutoff from a temporal perspective. However, the multimodal field still lacks benchmarks for evaluating the updating of existing knowledge, and the impact of knowledge injection on general capabilities as well as cross-modal trans-

fer has not yet been systematically analyzed.

### 2.3. Knowledge Injection

Knowledge injection aims to enable models to acquire and internalize knowledge. While retrieval-augmented generation (RAG) (Lewis et al., 2020; Oche et al., 2025) provides on-the-fly supplementation via external retrieval, it fails to persist knowledge in model parameters and suffers from high retrieval overhead at scale. Consequently, prior work has focused on parameter-level knowledge injection, including supervised fine-tuning and parameter-efficient variants such as LoRA (Hu et al., 2022; Dong et al., 2024; Jiang et al., 2025b), as well as preference-based methods such as RLHF and its variants (e.g., CPO, SimPO, and ORPO) (Xu et al., 2024a; Meng et al., 2024; Hong et al., 2024). To reduce fine-tuning cost and mitigate parameter interference, knowledge editing methods (e.g., GRACE and WISE) (Hartvigsen et al., 2023; Wang et al., 2024b) perform localized parameter updates or leverage external memory to modify specific facts.

## 3. MMKU-Bench

### 3.1. Problem Definition

This paper defines the following three primary problems in multimodal settings:

**Knowledge Updating.** Knowledge updating aims to correct outdated or incorrect knowledge stored in a model. Let the model's original knowledge be denoted as $K_{\text{old}}$ and the updated knowledge as $K_{\text{update}}$. Given an image $I$ and its associated query $Q$, the model updates the knowledge related to $K_{\text{old}}$ in the original parameters to new knowledge via the function $f(I, Q; K_{\text{update}})$. This task focuses on whether the

model can accurately update the relevant knowledge in its original parameters to the new knowledge without affecting other unrelated knowledge, while maintaining the stability of its original general capability.

**Unknown Knowledge Injection.** Unknown knowledge injection examines the model's ability to acquire new knowledge in the absence of prior information. Given an image $I$ and a query $Q$ involving unknown knowledge $K_{unknown}$, where $K_{unknown} \notin \mathcal{D}_{train}$, the model is required to learn the relevant knowledge through a knowledge injection mechanism and generate the correct target response $y_{target}$. This setting is mainly used to evaluate the model's ability for rapid learning and generalization to unknown knowledge.

**Cross-modal Transfer.** To evaluate the transfer and retention of model knowledge across different modalities, this paper defines three evaluation paradigms based on the modality combinations used at the knowledge injection and evaluation stages. Here, M denotes multimodal data and T denotes text data. Specifically, M → M, M → T, and T → M, corresponding to multimodal injection–multimodal evaluation, multimodal injection–text evaluation, and text injection–multimodal evaluation, respectively.

### 3.2. Benchmark Construction

To construct a high-quality benchmark for knowledge updating evaluation, we design a systematic pipeline for multimodal question answering data construction, and name the benchmark MMKU-Bench. The overall construction pipeline is illustrated in Figure 2.

**Step 1: Knowledge Collection.** We construct knowledge sources from two aspects: updated knowledge and unknown knowledge. For updated knowledge, we first select QA pairs from the EVQA dataset (Mensink et al., 2023) that can be correctly answered by both InternVL2.5-8B and Qwen2.5-VL-7B. We then employ ChatGPT-4o (Hurst et al., 2024) to perform counterfactual editing by generating new QA instances with entirely different answers while keeping the questions unchanged. During this process, a self-check mechanism is introduced to ensure answer diversity and strict adherence to predefined constraints. For unknown knowledge, we obtain local Wikipedia dumps dated February 1, 2025 and November 1, 2025, identify entities that differ between the two snapshots to extract newly introduced knowledge, and collect relevant images for these entities from Google and Bing to construct candidate image sets.

**Step 2: QA Generation and Paraphrasing.** Based on the collected knowledge, we use GPT-4o to generate initial QA pairs, with local Wikipedia dumps and online retrieval results serving as reliable evidence. The questions are then paraphrased to enhance linguistic diversity while preserving

their original semantics, resulting in two separate QA sets for knowledge injection and model evaluation, respectively. During VQA construction, we employ CLIP to compute cosine similarity between candidate images, discard samples with excessively high ($>0.9$) or low ($<0.7$) semantic similarity, and select two images that are semantically related yet visually distinct as the preliminary visual inputs.

**Step 3: Cross-modal Data Construction.** To enable cross-modal transfer comparison, we construct cross-modal evaluation data based on the original VQA samples. During the initial data collection process, the entity names corresponding to the knowledge and images are already available. Leveraging this information, we use GPT-4o to revise the previously generated QA pairs by replacing referential expressions such as "this game" and "the game" with the corresponding entity names (e.g., "Battlefield 6"). Based on this processing, we directly remove the images and retain only the textual data, thereby constructing datasets for text-based knowledge injection and text-based evaluation to support cross-modal transfer analysis.

**Step 4: Quality Control.** To ensure data quality, we conduct multiple rounds of verification on both QA and images, including manual review by three annotators. The review process focuses on the correctness of edited answers, the clarity of question formulations, and the consistency between images and questions. To improve efficiency, we develop a dedicated evaluation interface that clearly presents QA along with their corresponding images, facilitating fast and systematic human inspection.

### 3.3. Benchmark Analysis

*Table 1.* Key Statistics of MMKU-Bench.

| Statistic | Updated | Unknown |
|---|---|---|
| Knowledge | 13,483 | 12,309 |
| Subfields | 156 | 175 |
| Number of unique images | 25,077 | 24,618 |
| - Injection images | 12,506 | 12,309 |
| - Evaluation images | 12,571 | 12,309 |
| Updated knowledge length | | |
| - Maximum length | 31 | 28 |
| - Average length | 10.85 | 11.81 |
| Evaluation question length | | |
| - Maximum length | 50 | 39 |
| - Average length | 21.11 | 20.30 |
| Evaluation answer length | | |
| - Maximum length | 15 | 15 |
| - Average length | 1.51 | 2.19 |

As shown in Table 1, MMKU-Bench is a large-scale bench

*Table 2.* Performance comparison between InternVL2.5-8B and Qwen2.5-VL-7B.

| | | InternVL2.5-8B | | | | | Qwen2.5-VL-7B | | | | |
| | | Updated | | | Unknown | | Updated | | | Unknown | |
| Method | | Correct ↑ | F1 ↑ | Outdated ↓ | Correct ↑ | F1 ↑ | Correct ↑ | F1 ↑ | Outdated ↓ | Correct ↑ | F1 ↑ |
|---|---|---|---|---|---|---|---|---|---|---|---|
| **Baseline** | Vanilla | – | – | 100 | 5.03 | 11.97 | – | – | 100 | 5.17 | 11.09 |
| | Golden Context | 67.02 | 66.28 | 6.50 | 76.03 | 86.55 | 69.44 | 67.36 | 5.01 | 79.36 | 88.78 |
| **SFT** | Full-FT | **35.19** | **11.16** | 5.53 | 17.44 | 14.38 | **38.40** | **11.82** | 6.10 | 18.16 | 14.40 |
| | LoRA | 29.54 | 9.64 | 7.74 | 12.74 | 11.92 | 31.53 | 9.81 | 8.45 | 11.16 | 10.39 |
| **RLHF** | CPO | 33.35 | 10.57 | 4.00 | 16.76 | 14.04 | 36.65 | 11.25 | **4.73** | **18.47** | **14.68** |
| | ORPO | **35.19** | 11.25 | 4.85 | **17.65** | **14.42** | 33.56 | 10.41 | 6.03 | 15.08 | 12.43 |
| | SimPO | 33.18 | 10.49 | **3.94** | 17.39 | 14.26 | 36.70 | 11.37 | 4.88 | 18.19 | 14.63 |
| **KE** | GRACE | 0.61 | 3.97 | 30.18 | 0.28 | 10.00 | 0.74 | 4.37 | 31.85 | 0.31 | 9.55 |
| | WISE | 6.06 | 6.68 | 39.45 | 6.54 | 13.28 | 5.87 | 6.56 | 45.72 | 6.11 | 12.26 |

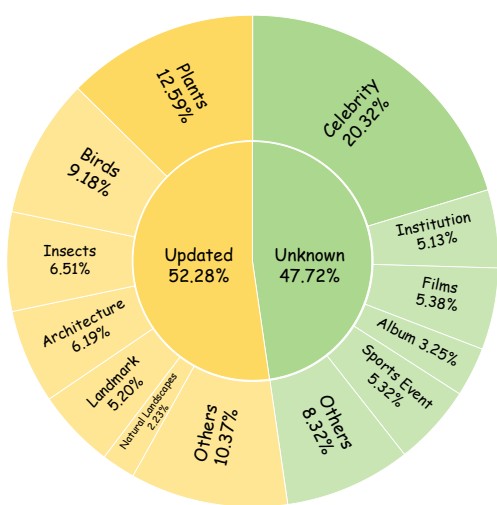

*Figure 3.* Category distribution of MMKU-Bench. "Others" represents categories not explicitly displayed.

comprising 25,792 knowledge entries and over 49k unique images to reduce visual overlap between the knowledge injection and evaluation phases. The evaluation primarily consists of descriptive questions with concise ground-truth answers to support objective assessment. Figure 3 shows a balanced distribution of the two knowledge categories (Updated: 52.28%, Unknown: 47.72%): Updated knowledge mainly covers natural domains such as plants and birds, while Unknown knowledge is dominated by celebrity-related content, followed by cultural topics including films and sports, ensuring both content diversity and real-world relevance.

## 4. Experiment

### 4.1. Experimental Setup

**Large Multimodal Models.** To ensure a fair and interpretable comparison, we select two representative LMMs

with synchronized release cycles and comparable parameter scales: **InternVL2.5-8B** (Dec. 2024) and **Qwen2.5-VL-7B** (Jan. 2025).

**Knowledge Injection Methods.** We evaluate RAG with Golden Context, SFT, RLHF, and Knowledge editing methods; details are provided in the appendix D.

**Evaluation Metrics.** To quantitatively evaluate model performance on QA tasks, we use two standard metrics: Cover Exact Match (CEM)(Xu et al., 2024b) and F1 score (F1)(He et al., 2025). Let the model output be denoted as $\hat{Y}$ and the ground-truth answer set as $Y$. The CEM is defined as CEM = 1 if $Y \subseteq \hat{Y}$, and CEM = 0 otherwise. For the updated tasks, we report CEM under two settings: Correct, computed on the updated answer, and Outdated, computed on the outdated answer.

For the F1 score, we first represent the ground-truth answers and model output as word sets $W(Y) = \{y_1, \ldots, y_m\}$ and $W(\hat{Y}) = \{\hat{y}_1, \ldots, \hat{y}_n\}$, respectively. The number of overlapping words is denoted by $U(\hat{Y}, Y) = \sum_{t \in W(Y)} 1[t \in W(\hat{Y})]$. Precision and recall are then given by $\mathcal{P}(\hat{Y}, Y) = \frac{U(\hat{Y}, Y)}{|W(\hat{Y})|}$ and $\mathcal{R}(\hat{Y}, Y) = \frac{U(\hat{Y}, Y)}{|W(Y)|}$, respectively. Finally, the F1 score is computed as the harmonic mean of precision and recall: $F1 = \frac{2 \cdot \mathcal{P} \cdot \mathcal{R}}{\mathcal{P} + \mathcal{R}}$.

### 4.2. Main Results

The experimental results of different knowledge injection strategies on the Updated and Unknown tasks are summarized in Table 2. Based on these results, we make the following observations:

**Obs 1: Even with sufficient context for RAG, LMMs remain limited in knowledge updating scenarios.** Despite being provided with sufficient and highly relevant external knowledge at inference time, models perform significantly worse on Updated knowledge tasks than on Unknown knowl-

*Table 3.* General capability results of InternVL 2.5. Red values denote degradation, with darker shades indicating greater severity.

| Method | Comprehensive | | OCR | | Multidisciplinary | | Instruction | Mathematical | | Hallucination | |
|---|---|---|---|---|---|---|---|---|---|---|---|
| | MME↑ | MMBench↑ | SEED$^{BP}$↑ | OCRBench↑ | ScienceQA↑ | MMMU↑ | MIA-Bench↑ | MathVista↑ | MathVision↑ | POPE↑ | HallusionBench↑ |
| Vanilla | 1688.2 | 82.4 | 69.3 | 88.7 | 98.4 | 53.6 | 81.8 | 68.9 | 27.3 | 89.6 | 67.9 |
| Full-FT$_{Updated}$ | 499.1 | 74.0 | 64.0 | 64.4 | 95.3 | 44.7 | 62.8 | 49.5 | 19.4 | 49.9 | 34.3 |
| Full-FT$_{Unknown}$ | 1090.6 | 76.1 | 66.4 | 69.1 | 97.5 | 52.1 | 72.7 | 59.7 | 20.4 | 82.9 | 59.8 |
| LoRA$_{Updated}$ | 984.7 | 73.6 | 63.9 | 80.3 | 97.5 | 45.3 | 74.1 | 60.4 | 22.4 | 56.5 | 63.6 |
| LoRA$_{Unknown}$ | 1515.0 | 75.5 | 66.1 | 75.6 | 97.8 | 49.3 | 77.6 | 63.6 | 19.7 | 86.6 | 63.7 |
| CPO$_{Updated}$ | 511.8 | 62.5 | 62.5 | 38.1 | 92.6 | 48.7 | 61.0 | 45.0 | 18.8 | 53.2 | 58.0 |
| CPO$_{Unknown}$ | 1119.9 | 71.7 | 65.2 | 64.3 | 96.9 | 48.3 | 70.0 | 54.4 | 22.4 | 81.4 | 57.9 |
| ORPO$_{Updated}$ | 551.9 | 71.7 | 66.0 | 60.6 | 94.0 | 50.3 | 62.3 | 45.8 | 19.1 | 49.5 | 56.3 |
| ORPO$_{Unknown}$ | 935.3 | 76.2 | 65.9 | 66.9 | 97.4 | 50.9 | 71.9 | 57.5 | 21.4 | 77.7 | 55.6 |
| SimPO$_{Updated}$ | 469.1 | 61.6 | 60.9 | 37.1 | 91.1 | 46.2 | 60.1 | 45.9 | 21.4 | 39.9 | 57.4 |
| SimPO$_{Unknown}$ | 1161.3 | 73.0 | 65.7 | 62.7 | 96.9 | 52.3 | 68.6 | 57.2 | 20.1 | 81.4 | 58.7 |
| WISE$_{Updated}$ | 1672.6 | 82.1 | 69.1 | 88.4 | 97.9 | 53.4 | 80.4 | 67.2 | 27.1 | 88.3 | 62.4 |
| WISE$_{Unknown}$ | 1671.2 | 82.3 | 68.7 | 88.1 | 98.4 | 53.5 | 81.8 | 68.3 | 27.4 | 87.6 | 65.8 |

*Table 4.* General capability results of Qwen2.5 VL. Red values denote degradation, with darker shades indicating greater severity.

| Method | Comprehensive | | OCR | | Multidisciplinary | | Instruction | Mathematical | | Hallucination | |
|---|---|---|---|---|---|---|---|---|---|---|---|
| | MME↑ | MMBench↑ | SEED$^{BP}$↑ | OCRBench↑ | ScienceQA↑ | MMMU↑ | MIA-Bench↑ | MathVista↑ | MathVision↑ | POPE↑ | HallusionBench↑ |
| Vanilla | 1691.6 | 82.2 | 69.5 | 88.4 | 88.7 | 52.6 | 80.4 | 69.1 | 26.3 | 86.4 | 65.2 |
| Full-FT$_{Updated}$ | 393.9 | 30.3 | 28.2 | 64.0 | 59.0 | 35.3 | 56.7 | 47.4 | 19.1 | 61.8 | 16.9 |
| Full-FT$_{Unknown}$ | 1162.7 | 72.5 | 41.9 | 75.9 | 83.3 | 47.3 | 69.3 | 58.7 | 19.1 | 75.6 | 28.2 |
| LoRA$_{Updated}$ | 1153.7 | 9.1 | 20.2 | 56.3 | 39.4 | 32.7 | 67.2 | 42.6 | 39.3 | 81.8 | 26.7 |
| LoRA$_{Unknown}$ | 1504.8 | 60.2 | 42.3 | 76.5 | 82.6 | 49.3 | 73.6 | 59.6 | 20.4 | 80.5 | 36.2 |
| CPO$_{Updated}$ | 299.6 | 15.2 | 19.9 | 26.5 | 48.9 | 34.7 | 51.8 | 45.6 | 16.1 | 38.6 | 13.6 |
| CPO$_{Unknown}$ | 1210.0 | 71.5 | 41.5 | 72.6 | 84.3 | 55.3 | 68.6 | 56.7 | 20.1 | 73.5 | 39.6 |
| ORPO$_{Updated}$ | 184.4 | 13.7 | 17.7 | 52.0 | 43.1 | 33.1 | 46.5 | 42.3 | 17.4 | 49.5 | 13.4 |
| ORPO$_{Unknown}$ | 632.4 | 68.6 | 40.1 | 73.9 | 82.5 | 47.9 | 60.8 | 57.2 | 20.7 | 65.4 | 22.9 |
| SimPO$_{Updated}$ | 365.9 | 12.5 | 20.1 | 30.5 | 50.6 | 36.7 | 49.3 | 42.0 | 18.4 | 23.2 | 20.2 |
| SimPO$_{Unknown}$ | 1176.1 | 70.1 | 37.1 | 69.8 | 79.5 | 64.5 | 64.4 | 54.0 | 18.4 | 74.4 | 39.6 |
| WISE$_{Updated}$ | 1685.4 | 82.2 | 69.9 | 88.3 | 88.4 | 52.6 | 81.3 | 67.0 | 27.6 | 87.1 | 59.6 |
| WISE$_{Unknown}$ | 1681.1 | 82.2 | 69.5 | 88.5 | 88.6 | 53.3 | 80.8 | 68.4 | 25.7 | 86.4 | 64.9 |

edge tasks. This suggests that when externally supplied knowledge conflicts with knowledge implicitly encoded in pretrained parameters, models struggle to effectively integrate new information.

**Obs 2: Models exhibit stronger adaptability when learning updated knowledge than learning unknown knowledge.** Compared with the introduction of entirely unknown knowledge, updating a model's existing knowledge generally demonstrates better adaptability. Such tasks involve revising attributes of existing entities or relational facts, enabling the model to leverage semantic and structural priors formed during pretraining, thereby reducing the difficulty of knowledge injection and optimization.

**Obs 3: RLHF can reduce the model's reliance on outdated knowledge to some extent.** Experimental results show that, compared with SFT, RLHF is more effective at weakening the model's reliance on outdated knowledge. However, this advantage is mainly reflected in reducing erroneous preferences; its improvements in overall knowledge injection effectiveness and final performance gains remain limited and unstable.

**Obs 4: Existing knowledge editing methods struggle to support large-scale, continual knowledge injection.** Existing knowledge editing methods perform poorly overall under our experimental setting, reflecting inherent limitations in their methodological design: these methods are typically optimized for isolated, small-scale editing tasks. When confronted with large-scale, continuous knowledge update demands, they fail to support the long-term evolution of large multimodal models in real-world application.

### 4.3. General Capability Evaluation

To systematically analyze the impact of different knowledge injection methods on the general capabilities of multimodal models, we evaluate models fine-tuned with representative methods on 11 benchmark datasets across six dimensions, including: (i) **Comprehensive**: MME (Fu et al., 2025a) and MMBench (Liu et al., 2024); (ii) **OCR**: SEEDBench2 Plus (Li et al., 2024a) and OCRBench (Liu et al., 2023b); (iii) **Multidisciplinary**: ScienceQA (Lu et al., 2022) and MMMU (Yue et al., 2024); (iv) **Instruction Following**: MIABench (Qian et al., 2024); (v) **Mathematical Reasoning**:

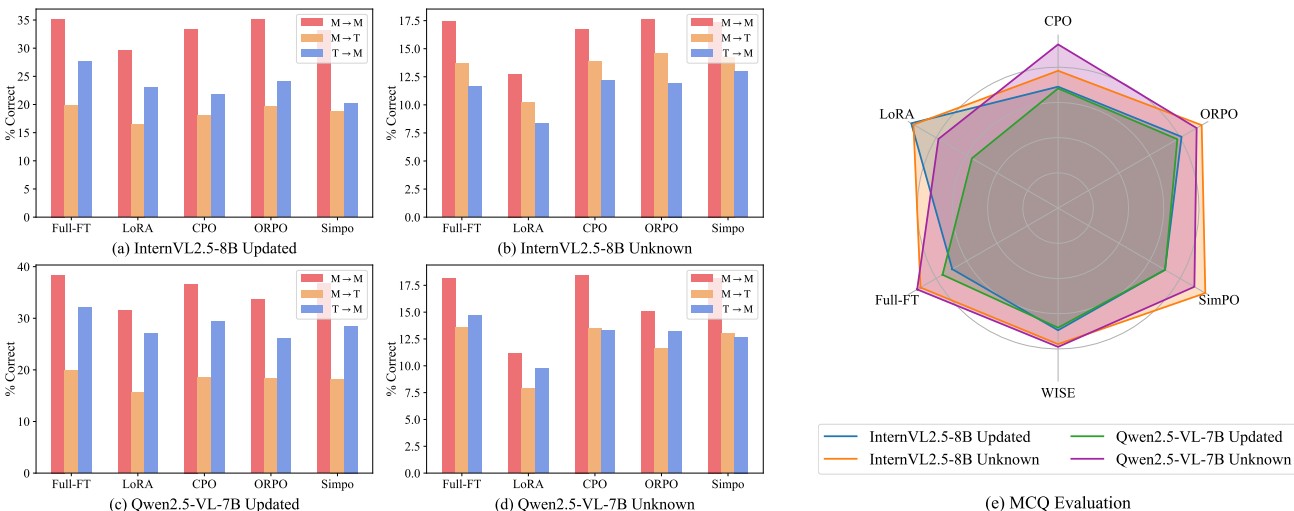

*Figure 4.* Consistency analysis of models trained with different methods. Left: Cross-modal transfer. Right: Decision consistency.

MathVista (Lu et al., 2023) and MathVision (Wang et al., 2024a); and (vi) **Hallucination**: POPE (Li et al., 2023b) and HallusionBench (Guan et al., 2024). Table 3 and Table 4 respectively present the general capability evaluation results of InternVL 2.5 and Qwen2.5 VL. Based on these evaluation results, we derive the following observations:

**Obs 5: General capability degradation is common after knowledge injection.** Experimental results show that, except for knowledge editing methods, most fine-tuning strategies lead to varying degrees of degradation in general capabilities across different evaluation dimensions after knowledge injection. This indicates that introducing new knowledge through direct parameter updates is often accompanied by notable side effects on general abilities.

**Obs 6: Learning updated knowledge is more destructive than unknown knowledge.** Compared to injecting entirely unknown knowledge (Unknown), updating existing knowledge (Updated) tends to result in more severe degradation of general capabilities. In updated knowledge scenarios, this degradation can be attributed to conflicts between newly injected knowledge and the model's pre-existing parametric knowledge, as well as the resulting representational restructuring, which leads to the observed performance decline.

**Obs 7: Parameter-efficient fine-tuning mitigates capability degradation to some extent.** Benefiting from its parameter-efficient update mechanism, LoRA exhibits better robustness than full fine-tuning across most general capability benchmarks, alleviating the interference of knowledge injection with the model's original capability structure to a certain degree.

**Obs 8: Knowledge editing methods are the most robust in preserving general capabilities.** Knowledge edit-

ing methods (e.g., WISE) achieve performance highly consistent with the original model across all general capability evaluations, demonstrating that localized and targeted knowledge updates are particularly effective at maintaining overall model capabilities.

**Obs 9: The degree of capability degradation varies significantly among different models.** Overall, InternVL 2.5 exhibits relatively limited capability degradation, with noticeable performance drops observed only on a small number of benchmarks, such as MME, OCRBench, MathVision, and POPE. In contrast, Qwen2.5 VL shows substantial degradation across all tasks, with particularly severe declines in the Comprehensive, OCR, and Hallucination dimensions. Notably, its maximum performance drop on the MME benchmark exceeds 89%, indicating that Qwen2.5 VL is more susceptible to the effects of knowledge injection.

### 4.4. Consistency Analysis

To evaluate the reliability of LLMs during knowledge injection, we designed analysis experiments along two dimensions: cross-modal transfer and decision consistency.

**Cross-modal Transfer:** As illustrated in Figure 4 (Left), we design three experimental configurations to analyze the robustness of cross-modal alignment after model fine-tuning:

- **M → M:** Both knowledge injection and evaluation are conducted in the multimodal setting, serving as a fully modality-aligned baseline. Under this configuration, both models achieve the best overall performance, indicating a stronger capability to learn and update knowledge when modality alignment is preserved, suggesting that modality alignment is crucial for effective

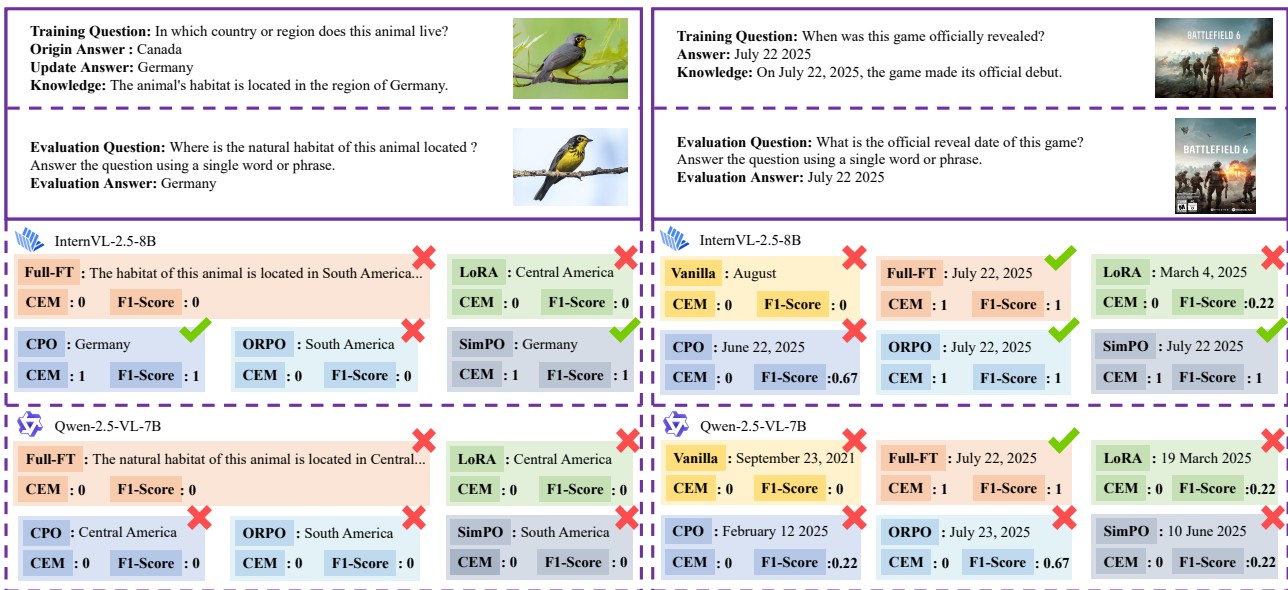

Figure 5. Case Study. Left: updated knowledge case. Right: unknown knowledge case.

knowledge updating in current LMMs.

- **M → T:** This configuration primarily examines whether the model can correctly modify textual knowledge based on image-grounded entities and their associated information. As shown in subfigures (a) and (c), for the knowledge updating task, the model struggles to effectively align image-based knowledge with the textual space, resulting in a substantial performance degradation. In contrast, for the novel knowledge task, where the target knowledge is unknown, multimodal inputs provide additional evidence, thereby alleviating the performance drop for both models.

- **T → M:** This represents another scenario of cross-modal transfer. Notably, for the knowledge updating task, text-based injection enables more direct modification of entity knowledge compared to multimodal injection, leading to better performance than M → T. However, on the unknown knowledge task, InternVL exhibits a clear performance decline, while QwenVL does not demonstrate a consistent advantage.

Overall, knowledge injected in a single modality does not readily generalize to other modalities, and cross-modal knowledge alignment remains a major challenge for LMMs.

**Decision Consistency:** We further examined the models' decision consistency through multiple-choice question (MCQ) experiments with distractors, conducted under multimodal settings. The results (see Figure 4, Right) show that accuracy on unknown knowledge is generally higher than that on updated knowledge, which is opposite to the phenomenon observed in open-ended question answering

tasks. This finding more clearly highlights the impact of semantic entanglement: in updated tasks, distractors are more likely to trigger residual memory biases from old knowledge, leading to shifts in decision-making; in the unknown scenario, the absence of explicit prior knowledge constraints allows the models to exhibit greater stability during decision-making.

## 5. Case Study

Figure 5 presents two representative case studies: the left illustrates a knowledge updating scenario involving a change in an animal's habitat, while the right depicts a new knowledge injection scenario concerning the announcement of a game's release date. Through qualitative analysis, we observe that during knowledge updating, models may sometimes produce erroneous outputs that are consistent with neither the original facts nor the updated evidence. For example, in the habitat updating case, although the habitat is updated from "Canada" to "Germany," several intervention methods still lead the model to generate irrelevant answers such as "South America," which are inconsistent with both the original and the updated facts.

In contrast, in the unknown knowledge scenario, when confronted with novel facts that did not appear in the training data, models often struggle to fully reproduce the correct answer. Despite the injected knowledge explicitly providing the complete date information, some models are only able to generate partial fragments or produce shifted dates. Moreover, strategies such as LoRA achieve extremely low F1 scores on both models, indicating a failure to accurately capture the complete newly injected fact.

# 6. Conclusion

This paper proposes MMKU-Bench, a benchmark to systematically evaluate multimodal knowledge updating. Based on this benchmark, we conduct an in-depth analysis of the behavioral differences exhibited by LMMs when updating existing knowledge versus learning unknown knowledge, revealing the limitations of current approaches in terms of knowledge injection, catastrophic forgetting, and cross-modal transfer. We expect MMKU-Bench to provide a unified evaluation standard for multimodal knowledge updating and to foster further progress in this direction.

Despite its advantages in terms of scale and data quality, MMKU-Bench still has several limitations. Due to the scarcity of real-world multimodal knowledge update data, the current knowledge updating process is constructed through counterfactual editing, which may introduce distribution shifts. Moreover, achieving low-cost and sustainable knowledge updating without compromising the overall capabilities of the model remains an open challenge. Future work will focus on developing evaluation benchmarks that more closely reflect real-world multimodal knowledge updates and on exploring more effective knowledge updating algorithms to address these challenges.

# Acknowledgments

This work is supported by the National Natural Science Foundation of China (62205181); the Shandong Province Outstanding Youth Science Fund Project (Overseas) (2023HWYQ-023); the Natural Science Foundation of Shandong Province (ZR2025QC1570). Fig 1 and 2 include icons adapted from resources provided by Flaticon.com.

# Impact Statement

This paper presents work whose goal is to advance the field of Machine Learning. There are many potential societal consequences of our work, none which we feel must be specifically highlighted here.

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

# A. Statistics of MMKU-Bench

## A.1. More Details on Fine-grained Types Distribution

*Table 5.* The fine-grained types in MMKU-Bench.

| Categories | Fine Grained Types |
|---|---|
| Updated | Common Plant, Bird, Tree, Flower, Church, Butterfly, Architectural Structure, Other, Castle, Moth, Shrub, Museum, Park, Beetle, Dragonfly, Wasp, Stadium, Bridge, Herb, Temple, Bee, Orchid, Fern, Monument, Residential Building, Lake, Mountain, Monastery, Palace, Cactus, Mosque, Square, Lighthouse, Coast, Skyscraper, Theater, Transportation Site, Hotel, Tower, Island, Insect Larvae, Moss and Lichen, Special Building, Historic Site, Fruit, Entertainment Site, Military Building, Reptile, Fly, Mosquito, Ocean, Ant, Tomb Architecture, Memorial Facility, Gate, Waterfall, Special Landform, Wilderness, Market Building, Protected Area, Industrial Site, Historic Area, Water Structure, Train Station, Synagogue, Library, Station Building, Ladybug, River, Fountain, Bird Extension, Amphitheater, Factory, School, Market, Building Component, Road Facility, Religious Site, Zoo, Beach Resort, Aqueduct, Fungus, Volcano, Tunnel, Cave, Amusement Park, Geographical Feature, Dam, Port Building, Road, Astronomical Facility, Port, Peninsula, Mollusk, Agricultural Site, Conference Facility, Fish, Human Related, Algae, Special Transportation, Fjord, Pavilion, Shopping Mall, Arthropod, City Wall, Special Plant Structure, Waterbody, Wetland, Specialized Plant, Mammal, Amphibian, Aquatic Mammal, Sports Facility, Desert, Cable Car, Ship, Cinema, Bell Tower, Colonnaded Building, Geothermal Feature, Natural Arch, Waterway Facility, Golf Course, Train, Valley, Transportation Hub, Aquatic Plant, Stone Formation, Public Building, Bat, Restaurant Building, Mangrove, Grasshopper, Crustacean, Wind Turbine, Airport, Race Track, Racecourse, Ski Resort, Campsite, Car, Pedestrian Street, Bus Station, Airport Building, Bicycle, Subway, Glacier, Coral Reef, Bus, Hot Air Balloon, Forest, Airplane, Parking Lot, Plain, Hospital, Worm |
| Unknown | Politician, Song, Musician, Company, Actor, Singer, Election, Book, Manga, Boxer, Anime, Soundtrack, Clergy, Award, Train, Tournament, Author, Car, Journalist, Lawyer, Smartphone, Government, Cricketer, Dinosaur, Festival, Software, Model, Athletics, Gymnast, Stadium, Aircraft, Protest, Diplomat, Ship, Warship, Music Album, Podcast, Wrestling, Bank, Documentary, Restaurant, Airline, Airport, Beverage, Brand, Bridge, Church, Concert, Hotel, Weightlifting, School, Video Game, Camera, Sculpture, Spacecraft, Weapon, Artificial Intelligence, Museum, Helicopter, Hospital, League, Motorcycle, Movie, Newspaper, Tablet, Tennis, Toy, Truck, Website, Computer, Cosmetics, Magazine, Mammal, Painting, Sports Team, Supermarket, TV Series, Airshow, Court, Doctor, Fictional Character, Food, Government Official, Jewelry, Plant, Router, Speaker, Sports Event, Weapon System, Aerospace, American Football Player, Animated Character, Architecture, Bakery, Baseball, Baseball Player, Basketball, Basketball Player, Bicycle, Bird, Board Game, Book Type, Cable Car, Car Manufacturer, Chess Player, Clothing, Consumer Electronics, Cycling Race, Election Results, Electronic Component, Electronics Manufacturer, Fashion Item, Film Director, Film Franchise, Film Studio, Film Type, Football, Football Kit, Football Player, Furniture, Game Character, Game DLC, Game Developer, Headphones, Historical Site, Hockey Player, Ice Cream, Ice Hockey, Language Vocabulary, Legal Case, Literature, Medical Equipment, Medical System, Military Base, Military Branch, Military Personnel, Military Vehicle, Mobile App, Motorsport Event, Mountain, Music Festival, Music Genre, Music Style, Non Fiction Book, Online Platform, Other, Payment Service, Payment System, Photography, Police, Political Party, Port, Printer, Race Track, Racing Driver, Radio Program, Retail Store, Road, Rugby Player, Shopping Mall, Snooker Player, Special Sport, Special Vehicle, Sports Season, Sports Team Season, TV Channel, TV Episode, TV Program, TV Season, Taxi, Tennis Player, Trade Center, Train Station, Volleyball, Volleyball Player |

Table 5 presents a comprehensive breakdown of the fine-grained type distribution in the MMKU-Bench benchmark, covering 156 distinct categories of updated knowledge and 175 distinct categories of unknown knowledge, which highlights the exceptional diversity of the benchmark. By jointly considering both updated and unknown knowledge, MMKU-Bench more faithfully reflects model performance in dynamic knowledge environments, providing a solid and reliable foundation for in-depth analysis, comparative evaluation, and future research directions.

## A.2. Word Cloud Distribution

In Figure 6(a), we present the word cloud distribution of the Updated Knowledge. It can be observed that words such as park, museum, and church appear more frequently, which may be attributed to the fact that the original EVQA dataset contains a large amount of data related to such types of buildings. Meanwhile, in Figure 6(b), we show the word cloud distribution of the Unknown Knowledge. Through the analysis of fine-grained subfield distributions, key statistics, word cloud distributions, and multiple perspectives, we comprehensively demonstrate the diversity of the MMKU-Bench benchmark.

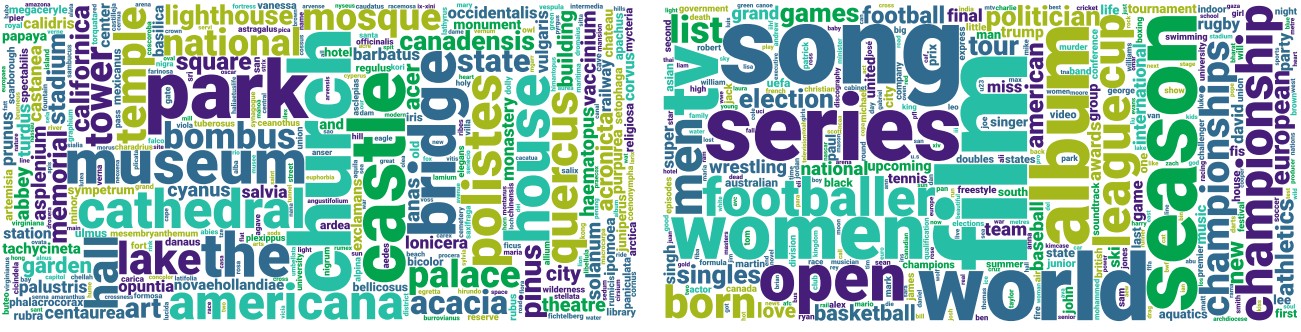

*(a)* Updated Knowledge                    *(b)* Upknown Knowledge

*Figure 6.* Word Cloud Distributions of MMKU-Bench

## B. More Results About MMKU-Bench

### B.1. The performance of consistency across different methods.

To systematically evaluate the consistency performance, we conducted extensive experiments, and the detailed results are summarized in Table 6.

*Table 6.* Cross-modal performance across different methods.

| Method | InternVL2.5-8B | | | | | Qwen2.5-VL-7B | | | | |
|---|---|---|---|---|---|---|---|---|---|---|
| | **Updated** | | | **Unknown** | | **Updated** | | | **Unknown** | |
| | **Correct ↑** | **F1 ↑** | **Outdated ↓** | **Correct ↑** | **F1 ↑** | **Correct ↑** | **F1 ↑** | **Outdated ↓** | **Correct ↑** | **F1 ↑** |
| Vanilla | - | - | 100 | 5.03 | 11.97 | - | - | 100 | 5.17 | 11.09 |
| RAG with Golden Context | 67.02 | 66.28 | 6.50 | 76.03 | 86.55 | 69.44 | 67.36 | 5.01 | 79.36 | 88.78 |
| Full-FT$_{M\rightarrow M}$ | 35.19 | 11.16 | 5.53 | 17.44 | 14.38 | 38.40 | 11.82 | 6.10 | 18.16 | 14.40 |
| Full-FT$_{M\rightarrow T}$ | 19.78 | 6.90 | 5.99 | 13.71 | 11.68 | 19.89 | 6.73 | 7.29 | 13.57 | 11.26 |
| Full-FT$_{T\rightarrow M}$ | 27.64 | 8.82 | 6.20 | 11.65 | 10.32 | 32.16 | 9.96 | 8.62 | 14.72 | 12.01 |
| LoRA$_{M\rightarrow M}$ | 29.54 | 9.64 | 7.74 | 12.74 | 11.92 | 31.53 | 9.81 | 8.45 | 11.16 | 10.39 |
| LoRA$_{M\rightarrow T}$ | 16.53 | 6.02 | 8.32 | 10.21 | 10.14 | 15.66 | 5.47 | 10.21 | 7.94 | 8.32 |
| LoRA$_{T\rightarrow M}$ | 23.08 | 7.81 | 8.82 | 8.37 | 9.02 | 26.98 | 8.51 | 9.63 | 9.72 | 9.41 |
| CPO$_{M\rightarrow M}$ | 33.35 | 10.57 | 4.00 | 16.76 | 14.04 | 36.65 | 11.25 | 4.73 | 18.47 | 14.68 |
| CPO$_{M\rightarrow T}$ | 18.03 | 6.30 | 4.65 | 13.84 | 11.78 | 18.52 | 6.29 | 5.37 | 13.48 | 11.16 |
| CPO$_{T\rightarrow M}$ | 21.81 | 6.88 | 4.78 | 12.17 | 10.91 | 29.41 | 9.22 | 7.04 | 13.28 | 11.18 |
| ORPO$_{M\rightarrow M}$ | 35.19 | 11.25 | 4.85 | 17.65 | 14.42 | 33.56 | 10.41 | 6.03 | 15.08 | 12.43 |
| ORPO$_{M\rightarrow T}$ | 19.74 | 6.86 | 5.58 | 14.61 | 12.06 | 18.40 | 6.21 | 6.57 | 11.59 | 9.67 |
| ORPO$_{T\rightarrow M}$ | 24.08 | 7.68 | 6.19 | 11.95 | 10.49 | 26.15 | 8.82 | 11.18 | 13.20 | 11.07 |
| SimPO$_{M\rightarrow M}$ | 33.18 | 10.49 | 3.94 | 17.39 | 14.26 | 36.70 | 11.37 | 4.88 | 18.19 | 14.63 |
| SimPO$_{M\rightarrow T}$ | 18.74 | 6.45 | 4.66 | 14.27 | 11.68 | 18.23 | 6.22 | 5.94 | 13.01 | 11.19 |
| SimPO$_{T\rightarrow M}$ | 20.17 | 6.56 | 4.73 | 12.96 | 11.27 | 28.52 | 8.90 | 6.70 | 12.69 | 10.90 |
| WISE$_{M\rightarrow M}$ | 6.06 | 6.68 | 39.45 | 6.54 | 13.28 | 5.87 | 6.56 | 45.72 | 6.11 | 12.26 |
| WISE$_{M\rightarrow T}$ | 4.37 | 4.74 | 42.18 | 5.02 | 12.31 | 5.22 | 5.81 | 48.42 | 5.48 | 11.63 |
| WISE$_{T\rightarrow M}$ | 5.68 | 6.12 | 40.76 | 5.37 | 12.89 | 5.64 | 6.02 | 46.38 | 5.92 | 12.08 |

We mainly compare six methods, including Full-FT, LoRA, as well as CPO, ORPO, SimPO, and WISE. Since GRACE is a memory-based method that does not introduce substantial modifications to model parameters and exhibits almost no generalization capability, it is excluded from the subsequent analysis. In addition, under our experimental settings, WISE

performs poorly. According to prior studies(Qi et al., 2024; Jia et al., 2025), existing knowledge editing methods are typically designed for single-edit scenarios, with the maximum number of supported consecutive edits usually not exceeding 3000. As a result, such methods fail to function effectively under our continuous knowledge injection setting.

Real-world knowledge is large-scale and predominantly expressed in natural language, whereas traditional knowledge editing methods are mostly designed based on structured triples. This mismatch leads to clear limitations when applying such methods to realistic knowledge injection and update scenarios. Consistent with the observations in Figure 4, all methods achieve relatively better performance under the setting of multimodal knowledge injection with multimodal evaluation. Notably, in the updated knowledge task, we require the model predictions to exactly match the original answers; therefore, the accuracy under the Outdated setting is 100%. In contrast, for the unknown knowledge task, we observe that the Vanilla model achieves non-zero performance, which can be attributed to model hallucination.

Based on the results reported in the table, we conclude that for multimodal large language models, cross-modal inconsistency remains the core challenge in current multimodal knowledge injection.

### B.2. Knowledge Injection Performance Across Categories

*Table 7.* Performance Comparison of Different Methods Across Categories. The yellow cells indicate the method achieving the best average performance, red cells denote the best-performing results within each sub-category, and blue cells represent the worst performance.

| Method | Updated Avg CEM↑ | Avg F1↑ | Biology CEM↑ | Biology F1↑ | Architecture CEM↑ | Architecture F1↑ | Landmark CEM↑ | Landmark F1↑ | Natural landscape CEM↑ | Natural landscape F1↑ | Unknown Avg CEM↑ | Avg F1↑ | Celebrity CEM↑ | Celebrity F1↑ | Film CEM↑ | Film F1↑ | Album CEM↑ | Album F1↑ | Sports Event CEM↑ | Sports Event F1↑ |
|---|---|---|---|---|---|---|---|---|---|---|---|---|---|---|---|---|---|---|---|---|
| *InternVL2.5-8B* | | | | | | | | | | | | | | | | | | | | |
| Vanilla | - | - | - | - | - | - | - | - | - | - | 5.03 | 11.97 | 5.04 | 11.17 | 9.85 | 12.43 | 6.11 | 12.65 | 5.30 | 11.93 |
| Full-FT | 35.19 | 11.16 | 32.66 | 10.15 | 42.86 | 13.66 | 40.15 | 13.64 | 35.07 | 11.15 | 17.44 | 14.38 | 16.87 | 13.11 | 16.13 | 13.28 | 17.04 | 15.83 | 15.23 | 15.77 |
| LoRA | 29.54 | 9.64 | 27.79 | 8.98 | 33.40 | 10.89 | 32.69 | 11.15 | 32.12 | 10.30 | 12.74 | 11.92 | 11.93 | 10.55 | 14.40 | 12.79 | 9.42 | 11.17 | 12.76 | 14.76 |
| CPO | 33.35 | 10.57 | 30.05 | 9.32 | 41.04 | 13.24 | 40.00 | 13.43 | 41.32 | 12.91 | 16.76 | 14.04 | 15.72 | 12.51 | 16.56 | 13.89 | 15.38 | 15.16 | 15.45 | 15.91 |
| ORPO | 35.19 | 11.25 | 31.98 | 10.09 | 43.17 | 13.76 | 42.24 | 14.18 | 41.32 | 13.09 | 17.65 | 14.42 | 16.49 | 12.45 | 16.45 | 13.86 | 16.21 | 15.11 | 15.82 | 17.05 |
| SimPO | 33.18 | 10.49 | 30.20 | 9.41 | 39.97 | 12.51 | 40.60 | 13.57 | 38.37 | 12.19 | 17.39 | 14.26 | 12.86 | 11.19 | 18.78 | 17.08 | 16.81 | 12.68 | 16.33 | 11.97 |
| WISE | 6.06 | 6.68 | 6.68 | 7.43 | 5.73 | 6.61 | 6.33 | 7.33 | 5.26 | 6.30 | 6.54 | 13.28 | 5.52 | 10.24 | 10.36 | 18.74 | 6.55 | 13.98 | 5.55 | 13.47 |
| *Qwen2.5-VL-7B* | | | | | | | | | | | | | | | | | | | | |
| Vanilla | - | - | - | - | - | - | - | - | - | - | 5.17 | 11.09 | 5.91 | 9.72 | 10.63 | 16.34 | 6.17 | 10.66 | 6.35 | 15.18 |
| Full-FT | 38.40 | 11.82 | 35.32 | 10.63 | 46.30 | 14.82 | 43.36 | 14.32 | 43.06 | 13.27 | 18.16 | 14.40 | 15.47 | 11.82 | 18.14 | 14.43 | 17.88 | 15.78 | 18.51 | 17.77 |
| LoRA | 31.53 | 9.81 | 29.47 | 8.97 | 37.03 | 11.73 | 35.00 | 11.78 | 32.29 | 9.97 | 11.16 | 10.39 | 9.92 | 8.61 | 12.17 | 11.97 | 10.61 | 10.81 | 12.17 | 11.97 |
| CPO | 36.65 | 11.25 | 33.37 | 9.98 | 44.80 | 14.24 | 42.84 | 14.09 | 39.41 | 12.14 | 18.47 | 14.68 | 15.21 | 11.58 | 18.14 | 15.20 | 18.83 | 17.52 | 20.48 | 18.37 |
| ORPO | 33.56 | 10.41 | 30.83 | 9.27 | 39.29 | 12.79 | 39.10 | 13.26 | 36.98 | 11.65 | 15.08 | 12.43 | 14.20 | 11.52 | 13.25 | 10.80 | 11.92 | 10.23 | 13.70 | 14.39 |
| SimPO | 36.70 | 11.37 | 34.01 | 10.31 | 43.61 | 13.86 | 40.97 | 13.69 | 38.89 | 11.91 | 18.19 | 14.63 | 13.46 | 11.04 | 21.17 | 14.18 | 16.66 | 12.58 | 13.41 | 9.26 |
| WISE | 5.87 | 6.56 | 6.35 | 7.04 | 5.50 | 5.71 | 4.93 | 5.85 | 4.34 | 4.87 | 6.11 | 12.26 | 6.16 | 12.04 | 9.81 | 17.28 | 6.08 | 11.19 | 5.79 | 10.40 |

Table 7 compares different knowledge injection methods across the Updated and Unknown categories and their corresponding sub-tasks. Overall, full-parameter fine-tuning (Full-FT) and preference-based optimization methods (e.g., ORPO and CPO) achieve more stable and superior performance on most updated knowledge tasks. In particular, on tasks such as Landmark and Natural Landscape, these methods significantly outperform parameter-efficient approaches in terms of CEM and F1, indicating that full-parameter optimization remains the most effective strategy for knowledge updating.

Within the Updated category, ORPO attains the best or second-best performance on multiple sub-tasks, demonstrating a favorable balance between knowledge injection strength and generation quality. By contrast, LoRA and SimPO, while competitive on some tasks, still exhibit a noticeable performance gap in overall effectiveness.

In the Unknown category, performance disparities become more pronounced. ORPO and CPO achieve the strongest average results across both model backbones, which can be attributed to their explicit treatment of hallucinated or incorrect outputs as rejected samples during training, enabling better discrimination of unknown knowledge. In contrast, WISE underperforms the original model in several sub-categories, suggesting that knowledge editing methods are not well suited for learning unknown knowledge.

## C. Qualitative Examples

In this section, we present four experimental settings corresponding to different data formats and evaluation protocols: multimodal injection with multimodal evaluation (Figure 7), multimodal injection with textual evaluation (Figure 8), textual injection with multimodal evaluation (Figure 9), and a multiple-choice question evaluation (Figure 10).

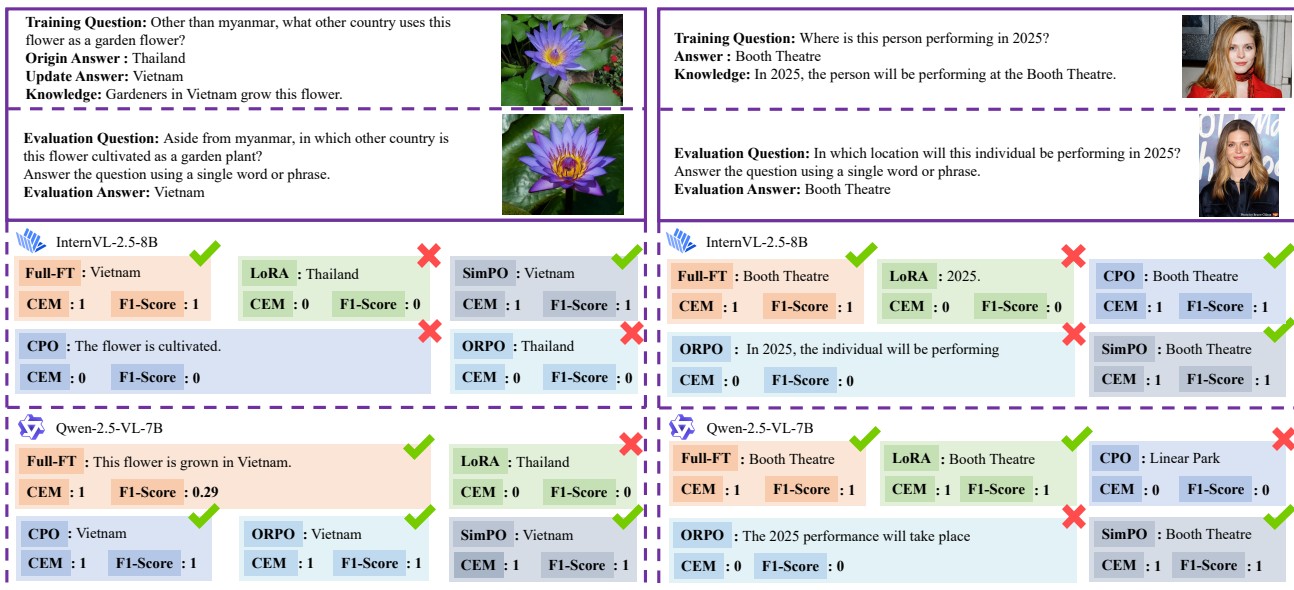

*Figure 7.* Case Study: Multimodal Knowledge Injection with Multimodal Evaluation

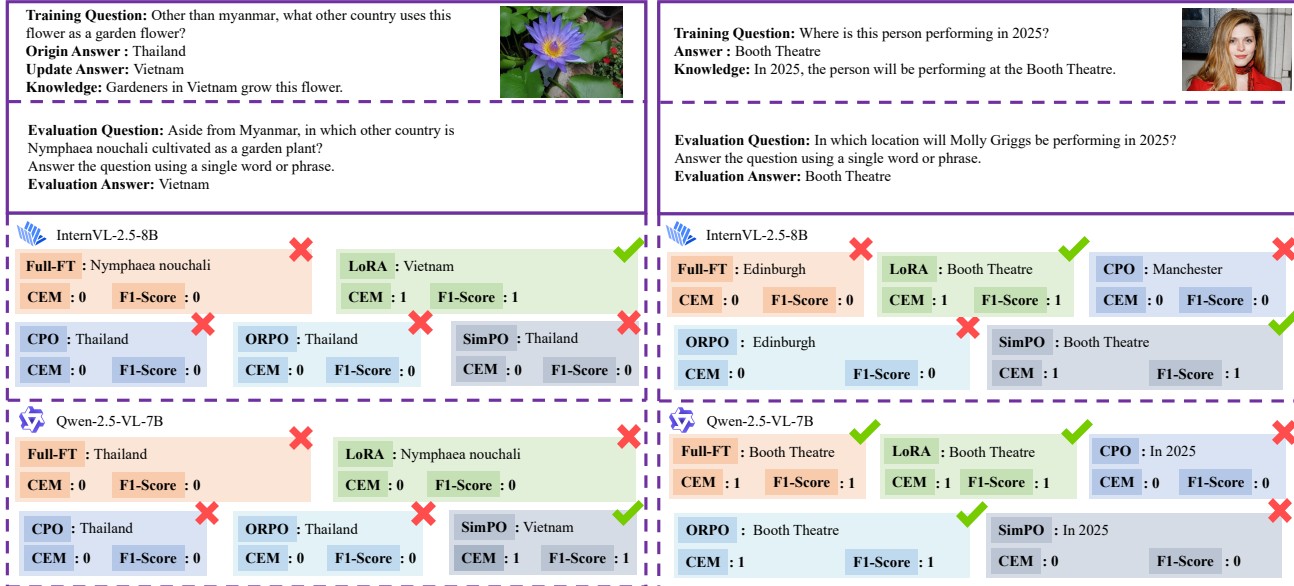

*Figure 8.* Case Study: Multimodal Knowledge Injection with Text-based Evaluation

In Figure 7, we replace the original entity names with their corresponding hypernyms during the QA generation process. Specifically, Nymphaea nouchali is referred to as this flower. Similarly, in the unknown-knowledge setting, Molly Griggs is replaced with this person. In Figure 8, to construct the textual evaluation setting, we remove the image input and replace the hypernyms with the original entity names. This design aims to evaluate whether the model can correctly align updated or injected knowledge to the textual modality under multimodal update and injection settings.

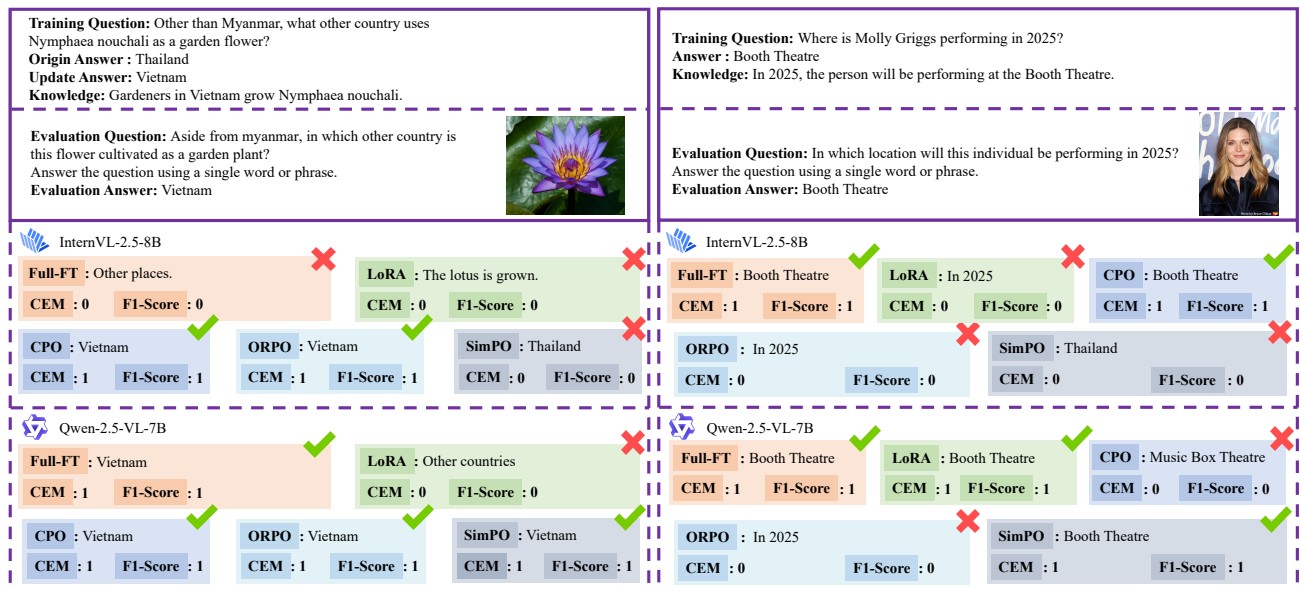

*Figure 9.* Case Study: Text-based Knowledge Injection with Multimodal Evaluation

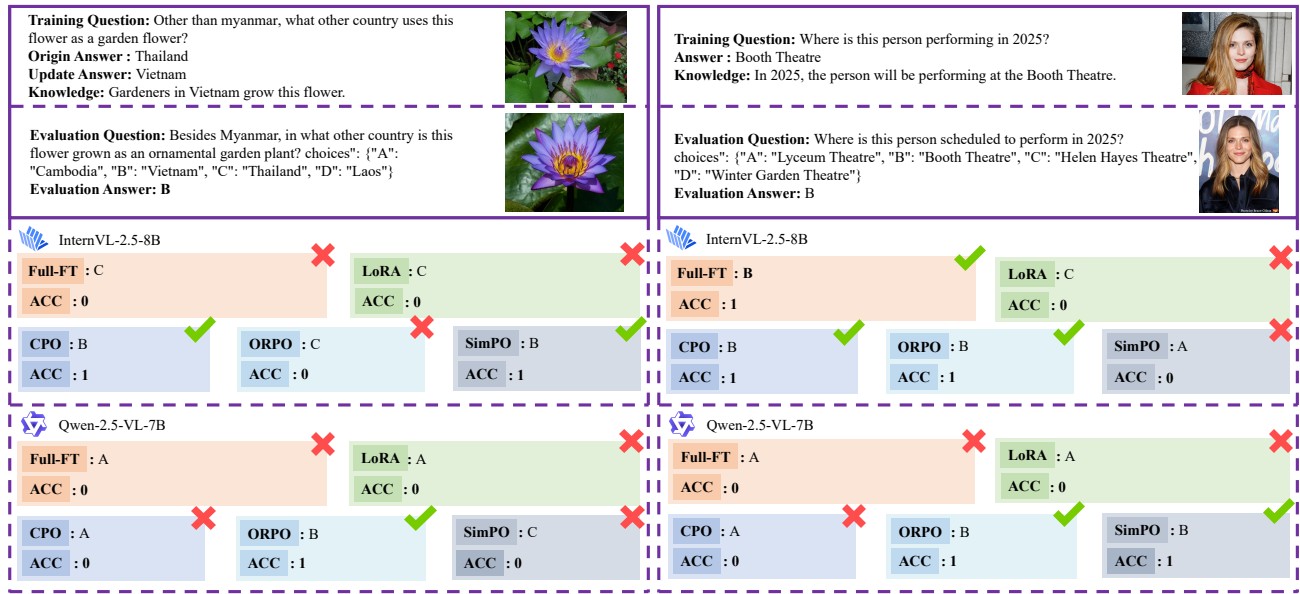

*Figure 10.* Case Study: Evaluation on Multiple-choice Question Answering

In contrast to Figure 8, Figure 9 converts multimodal knowledge into a textual form while retaining multimodal evaluation. Text-based injection provides a more direct supervision signal, resulting in a larger number of correct predictions for both updated and unknown knowledge. In Figure 10, we preserve the multimodal setting and focus on evaluating the model's decision consistency in the presence of multiple-choice options. For updated knowledge, the coexistence of outdated knowledge and additional distractor options introduces substantial difficulty for the model. In contrast, for unknown knowledge, the target entity does not exhibit strong associations with other options, which leads to better performance compared to the updated-knowledge setting.

## D. Detailed Experimental Settings

We conduct experiments using the MS-Swift[1] library, which supports a variety of large multimodal models and common fine-tuning methods. For knowledge editing, we employ the EasyEdit library[2]. To evaluate general capabilities, we adopt VLMEvalKit[3]. All experiments are carried out on four NVIDIA H200 GPUs with 141 GB memory each. In this study, we evaluate the following methods:

- **RAG with Golden Context**: This approach provides the model with gold-standard contexts containing all necessary information. It requires no parameter updates and serves as an empirical upper bound under ideal information retrieval conditions.

- **Supervised Fine-Tuning (SFT)**: SFT encourages models to internalize facts via further training on annotated data, enabling autonomous response generation without external aids. We implement two strategies: Full-FT and LoRA.

- **Reinforcement Learning from Human Feedback (RLHF)**: We align model knowledge through preference learning by constructing pairs where ground-truth facts are chosen, while outdated facts or erroneous outputs are rejected. We employ three reward-model-free variants: CPO, ORPO, and SimPO. In this study, all three methods are conducted under a full-parameter fine-tuning setting.

- **Knowledge Editing (KE)**: This paradigm focuses on parameter-efficient and locally controllable updates. We adopt two state-of-the-art methods: GRACE, which utilizes retrieval-driven memory editing, and WISE, which employs a dual-parameter memory mechanism for precise factual adaptation.

For SFT and RLHF, we set the number of training epochs to 6 and the batch size to 16. The learning rate is set to 1e-5 for full fine-tuning and 1e-4 for LoRA-based fine-tuning, with lora_rank set to 64 and lora_alpha set to 128. For the GRACE method, we perform edits on `language_model.layers[18]`, with the editing learning rate set to 1.0 and the number of editing iterations set to 20. For the WISE method, we conduct edits on `language_model.layers[23]`, using an editing learning rate of 1.0 and 10 editing iterations.

## E. Prompts for Different Steps



**Task Description.**
You are an expert question–answer editor. Given a question and its associated original answer, your task is to generate an alternative answer that adheres to the following constraints:

- - - - - - - - - - - - - - - - - - - - - - - - - - - - - -

**Rule1:** The generated answer must be different from the original answer.
**Rule2:** The generated answer must not be a synonym, paraphrase, or equivalent expression of the original answer (e.g., England vs. United Kingdom).
**Rule3:** The generated answer must belong to the same semantic category as the original answer.
**Rule4:** The answer must be concise, consisting of 1–5 words.
**Rule5:** Output only the answer text, without quotation marks, structured formatting, or explanatory content.

- - - - - - - - - - - - - - - - - - - - - - - - - - - - - -

**Input.**
Question: {question_text}
Forbidden answers: {forbidden_str}

**Output.**
A valid alternative answer satisfying all constraints above.



*Figure 11.* Prompt for updated knowledge editing.

---

[1] https://github.com/modelscope/ms-swift
[2] https://github.com/zjunlp/EasyEdit
[3] https://github.com/open-compass/VLMEvalKit

**Task Description.**
You are an academic knowledge editing assistant, tasked with rigorous counterfactual verification and rewriting of answers.

---

**Rule 1:** If the edited answer is semantically equivalent to or a paraphrase of the original answer, generate a completely new answer to the original question that is counterfactual and does not overlap in meaning with the original answer.
**Rule 2:** If the edited answer is not semantically equivalent to the original answer, return the edited answer unchanged.
**Rule 3:** Output only the final answer text. Do not include any explanations, justifications, or comments.

---

**Input.**
Original question: {text}
Original answer: {answer_original}
Edited answer: {answer_edited}

**Output.**
Strictly follow the rules when generating the answer.

*Figure 12.* Prompt for self-checking and verification.

**Task Description.**
You are a professional academic assistant specialized in precise question rephrasing.

---

**Rule 1:** The rewritten question must preserve exactly the same meaning as the original question.
**Rule 2:** The wording, sentence structure, and overall phrasing must differ substantially from the original question.
**Rule 3:** Simple synonym substitution is insufficient; the rewritten question must read as a natural and fluent alternative formulation.
**Rule 4:** Do not add, remove, or infer any information; the factual content must remain strictly identical.

---

**Input.**
Original question: {question_part}

**Output.**
Provide only the rewritten question, without any explanations or additional commentary.

*Figure 13.* Prompt for paraphrasing.

**Task Description.**
You are an expert academic assistant tasked with rewriting questions into standard knowledge-based question–answer form.

----------------------------------------

**Rule 1:** Replace vague visual references such as "this animal," "this plant," "this object," "this species," or similar expressions with the provided Wikipedia topic title.
**Rule 2:** Remove all mentions of visual cues, including references such as "in the image," "shown above," or "pictured here."
**Rule 3:** The rewritten question must remain semantically equivalent to the original question.
**Rule 4:** Do not invent, infer, or add any information that is not explicitly contained in the original question.
**Rule 5:** Use clear, fluent, and natural language appropriate for a standard knowledge-based QA setting.

----------------------------------------

**Input.**
Wikipedia topic: {wiki_name}
Original question: {question}

**Output.**
Provide only the rewritten question.

*Figure 14.* Prompt for entity replacement.

**Task Description.**
You are generating multiple-choice distractor answers for a given question.

----------------------------------------

**Rule 1:** Generate exactly {num_distractors} distractor answers.
**Rule 2:** All distractors must belong to the same semantic category as the correct answer.
**Rule 3:** All distractors must be incorrect for the given question.
**Rule 4:** Do not include the correct answer.
**Rule 5:** Do not include synonyms or paraphrases of the correct answer.
**Rule 6:** Output only a JSON list of strings, with no additional text.

----------------------------------------

**Input.**
Question: {question}
Correct answer: {correct_answer}
Category: {category}

**Output.**
Provide a JSON list containing exactly {num_distractors} distractor strings.

*Figure 15.* Prompt for multiple-choice question construction.

**Task Description.**
You are an expert in visual question–answer generation. Given a factual statement and a title describing the main subject of an image, your task is to generate a single high-quality visual question and its concise answer.

--------------------------------------------------------------------------------

**Rule 1:** Use the title only to infer the semantic type of the main entity (e.g., person, animal, object, building), and do not repeat or paraphrase the entity's real name in either the question or the answer.
**Rule 2:** When referring to the main subject in the question, always use an appropriate hypernym such as "this person," "this animal," "this building," or "this object," consistent with the semantic category implied by the title.
**Rule 3:** The question must be directly based on the given fact and describe a visually plausible attribute, action, or relationship, expressed using the hypernym rather than the entity name.
**Rule 4:** The answer must be concise and natural, consist of 1–4 words, contain no punctuation, and must not include the original entity name.
**Rule 5:** Generate both the question and the answer in English.
**Rule 6:** Output only a valid JSON object in the specified format, without any additional text, explanation, or formatting.

--------------------------------------------------------------------------------

**Examples.**
Title: Maine Coon Cat
Fact: In 2025, the Maine Coon became the most registered cat breed in the Cat Enthusiasts Association, surpassing the Ragdoll.
Output:
```
{
  "Question": "Which breed did this animal surpass to become the most registered breed?",
  "Answer": "Ragdoll"
}
```

--------------------------------------------------------------------------------

**Input.**
Title: {title}
Fact: {content}

**Output.**
Provide a JSON object in the following format:
```
{
  "Question": "question text",
  "Answer": "answer text"
}
```

*Figure 16.* Prompt for question–answer generation.

