# OpenReview forum: "MMKU-Bench: A Multimodal Update Benchmark for Diverse Visual Knowledge"
_ICML.cc/2026/Conference — ICML 2026 regular_

### Official Review · Reviewer_UAsc · 2026-03-08

**Soundness:** 4
**Presentation:** 3
**Significance:** 4
**Originality:** 4
**Overall Recommendation:** 5
**Confidence:** 4

**Summary:**

This paper introduces MMKU-Bench, a benchmark for evaluating knowledge updating in large multimodal models. The benchmark contains over 25k knowledge instances and more than 49k images, and focuses on two scenarios: Updated Knowledge, which introduces counterfactual edits to simulate outdated knowledge, and Unknown Knowledge, which focuses on previously unseen facts. The authors further evaluate several knowledge injection methods and analyse their effectiveness, cross-modal consistency, and robustness.

**Compliance With Llm Reviewing Policy:**

Affirmed.

**Final Justification:**

The authors have addressed all my questions in the rebuttal. I have no further concerns and am happy to maintain my score.

**Key Questions For Authors:**

1. Annotation Quality: Could you provide quantitative agreement metrics (e.g., IAA) for the manual review process to substantiate the quality of the ground truth?
2. Baseline Outdated Rates: The Vanilla Baseline exhibits a 100% "Outdated" rate. Is this strict adherence to outdated facts an absolute parametric limitation, or could it be mitigated via few-shot in-context learning?
3. Cross-Modal Shortcuts: In Step 3, removing images effectively reduces the multimodal task to unimodal text-to-text generation. How do you ensure the models are genuinely aligning updated text with visual features, rather than just exploiting text-based shortcuts?

**Limitations:**

yes

**Strengths And Weaknesses:**

Strengths
1. Novel Problem Scoping: The explicit decoupling of "learning unknown knowledge" from "updating outdated knowledge" is a highly valuable and timely contribution to the field of multimodal continual learning.
2. Rigorous Data Curation Pipeline: The methodology for constructing the benchmark is robust, notably employing CLIP for semantic similarity filtering to ensure image pairs are relevant yet visually distinct, alongside self-check mechanisms.
3. Cross-Modal Consistency Evaluation: By integrating text-only equivalents of the VQA tasks, the benchmark cleverly isolates visual recognition failures from fundamental parametric knowledge conflicts.
4. Insightful Empirical Findings: The comprehensive evaluation of modern LMMs (InternVL2.5, Qwen2.5-VL) exposes severe catastrophic forgetting in standard SFT and RLHF methods, providing a clear roadmap for future research.

Weaknesses
1. Missing Quality Assurance Metrics: The paper relies on a "manual review by three annotators" but lacks standard quantitative metrics (e.g., Inter-Annotator Agreement, Cohen’s/Fleiss' Kappa) to validate the dataset's reliability.
2. Risk of LLM Artifacts: Because the "Updated Knowledge" heavily relies on GPT-4o for counterfactual generation, the dataset may inadvertently favor models aligned with GPT-4o's specific lexical patterns or biases.
3. Superficial Failure Analysis: While the paper successfully demonstrates the catastrophic failure of SFT/RLHF compared to RAG, it does not sufficiently analyze the root cause (e.g., architectural bottlenecks vs. suboptimal hyperparameter tuning during knowledge injection).

---

> ### Author Rebuttal · Authors · 2026-03-31
>
> ## Dear Reviewer UAsc,
>
> > **W1&Q1: Missing Quality Assurance Metrics.**
>
> We filter the data from three aspects: **knowledge validity, language quality, and image quality**. Each sample is independently evaluated by three annotators, and only those with unanimous agreement and full approval are retained (e.g., Unknown is reduced from 20K to 12.3K):
>
> To quantitatively verify reliability, we randomly sample 50 instances and have them evaluated by three annotators.
> Among the 50 samples:
> - **47 samples** achieve complete agreement: including 32 unanimously accepted and 15 unanimously rejected.
> - **3 samples** show minor disagreement: all are cases where 2 annotators accepted and 1 rejected.
>
> We compute Fleiss' Kappa as:
> $$\kappa = \frac{0.96 - 0.5648}{1 - 0.5648} \approx \mathbf{0.908}$$
>
>
> > **W2: Risk of LLM Artifacts.**
>
>
> We address the issue of LLM artifacts through the following measures:
> - We maximize linguistic diversity and eliminate rigid patterns through multiple rounds of iteration and filtering.
> - In addition to verifying factual accuracy, human reviewers explicitly **filter out templated or unnatural expressions.
> - Crucially, MMKU focuses on **objective correctness** in VQA and MCQ tasks. Since success depends on domain knowledge, merely mimicking specific styles does not improve performance.
>
> > **W3: Insufficient analysis of the root causes behind SFT/RLHF failures.**
>
> We argue that the limitations of SFT/RLHF in knowledge updating mainly stem from:
>
> - **Parameter interference:** New knowledge may conflict with existing knowledge, leading to “knowledge inertia”;
> - **Semantic entanglement:** Local updates may perturb related semantic neighborhoods, affecting global stability;
> - **Objective mismatch:** The optimization objective is not tailored for precise overwriting of outdated knowledge, leading to over- or under-updating.
>
>
> In contrast, **RAG avoids modifying model parameters** through external retrieval, thus providing **greater stability**. In our experiments, we adopt common settings and perform basic tuning. Nevertheless, the **performance trends** across Updated and Unknown tasks **remain consistent and significant** across methods, indicating that the observed phenomenon is **not primarily driven by hyperparameters**.
>
> We acknowledge the importance of **failure analysis** and will further explore in future work:
>
> - Analyzing how different updating strategies affect the parameter space;
> - Exploring more stable knowledge updating methods;
> - Investigating ways to mitigate semantic entanglement and catastrophic forgetting.
>
>
> > **Q2: Is the 100% outdated rate due to parametric limitations or can it be mitigated by in-context learning?**
>
> Since the pretraining data of multimodal models is typically not publicly available, it is difficult to determine whether a model has encountered specific knowledge. Therefore, MMKU-Bench only retains **“outdated knowledge” that models can answer correctly and consistently**, ensuring that such knowledge has been internalized; otherwise, it would be impossible to distinguish whether failure is due to **“lack of updating” or “lack of prior knowledge.”** This design helpseliminate confounding factors and focuses on **the dynamic updating of known knowledge**.
>
> Meanwhile, we introduce **Golden Context** (see Table 2 in the paper) as a control. The results show that when explicit reference information is provided, models can **significantly adjust their outputs** and **reduce reliance on outdated knowledge**, indicating that their behavior is **not entirely constrained by parametric memory**.
>
> > **Q3: How do you ensure models rely on visual information rather than exploiting text-only shortcuts?**
>
> We have specifically designed our data construction process to **mitigate the issue of textual shortcuts** through the following mechanisms:
>
> - **Eliminating Direct Information Leakage:** During the multimodal input stage, we **anonymize visual entities**. By extracting an entity’s hypernym, we replace original names with **referential expressions** such as _"the person"_ or _"this game."_ This effectively prevents direct leakage via text, forcing the model to:
>
>     1. **Accurately identify** the specific entity from the image;
>
>     2. **Align** the updated knowledge with that corresponding visual entity.
>
> - **Validating Cross-Modal Generalization:** In the text-only evaluation stage, we remove the image and **restore referential expressions** to their original entity names. This requires the model to **internalize** the entity knowledge acquired from the visual modality and **generalize** it to pure textual reasoning.
>
> Under this design, the model cannot rely on shallow textual patterns. Instead, it must acquire core entities through visual information, thereby achieving genuine **cross-modal knowledge alignment and transfer**.

---

> > ### Author Rebuttal · Reviewer_UAsc · 2026-04-03
> >
> > The authors have addressed all my questions in the rebuttal. I have no further concerns and am happy to maintain my score.

---

> > > ### Author Response · Authors · 2026-04-07
> > >
> > > > **Dear Reviewer UAsc,**
> > >
> > > We are delighted that our response has addressed your concerns. We sincerely appreciate your encouraging feedback, especially your valuable comments regarding **quality control, parameter limitations, and cross-modal shortcuts.** We have revised the manuscript to further improve the scientific rigor of MMKU:
> > >
> > > 1. **Strict quality control**: The annotation protocol of MMKU and the Fleiss' Kappa results have been added to the appendix. We emphasize that only samples with complete agreement among all annotators are retained.
> > >
> > > 2. **Diverse knowledge generation**: We clarify that through multi-round iteration and manual review, we minimize templated or unnatural expressions introduced by GPT-4o as much as possible.
> > >
> > > 3. **Failure of parametric updating**: We highlight the limitations of parametric updating methods, including parameter interference, semantic entanglement, and constraints in optimization strategies.
> > >
> > > 4. **Avoiding cross-modal shortcuts**: We emphasize the rationale and role of hypernym substitution in MMKU dataset construction, which fundamentally prevents models from relying solely on textual shortcuts.
> > >
> > > If you have any further questions, we are happy to provide patient and detailed responses.
> > > Thank you again for your time, support, and constructive feedback.
> > >
> > > Sincerely,
> > > **Authors of Paper 19771**

---

### Official Review · Reviewer_ZZb7 · 2026-03-08

**Soundness:** 2
**Presentation:** 2
**Significance:** 3
**Originality:** 3
**Overall Recommendation:** 4
**Confidence:** 5

**Summary:**

This paper introduces MMKU-Bench, a benchmark designed for evaluating multi-modal knowledge update. It assesses a range of representative approaches, including supervised fine-tuning (SFT), reinforcement learning from human feedback (RLHF), and knowledge editing (KE).

**Compliance With Llm Reviewing Policy:**

Affirmed.

**Final Justification:**

The authors addressed my concerns about experimental settings, so I am inclined to maintain a positive score. However, their overclaim regarding theoretical contributions leads me to believe that the paper still requires polishing, even substantial revision. Therefore, I lean towards borderline accept.

**Key Questions For Authors:**

**Questions:**

1. While research on knowledge update is indeed important, given that real-world knowledge is continuously evolving and dynamic, I wonder whether a static benchmark can adequately capture and evaluate this characteristic.
2. I would be interested in seeing further analysis on how the knowledge already embedded within large language models (LLMs) contributes to or affects multi-modal knowledge update performance.

**Limitations:**

My primary concerns center on the quality control of the dataset, as well as the comprehensiveness of the methods evaluated. However, I would be willing to reconsider my assessment if the authors can more thoroughly address these concerns through additional data and analysis.

**Strengths And Weaknesses:**

**Strengths:**

1. The paper addresses an important research problem, particularly relevant to areas such as multi-modal continual learning, and provides a comparative evaluation of several baseline methods on the proposed benchmark.
2. The visualizations are well-designed and effectively support reader comprehension.

**Weaknesses:**

1. The introduction and comparison of the dataset are currently insufficient. The authors should provide more extensive data to highlight the distinctions and key improvements over existing datasets. Furthermore, additional evidence is needed to support the dataset's quality control, as GPT-4o may still produce errors or outdated knowledge.
2. The range of methods evaluated on the benchmark appears limited. I would suggest including models of varying sizes and, where feasible, closed-source APIs—even if their parameters cannot be updated—as they could still offer valuable reference points.
3. Although the paper is logically structured, certain expressions are ambiguous. For instance, the claim in the contributions section about “offering theoretical and experimental guidance” lacks sufficient theoretical support in the subsequent content.

---

> ### Author Rebuttal · Authors · 2026-03-31
>
> ## Dear Reviewer ZZb7,
> > **W1: Insufficient dataset comparison and quality validation.**
>
> MMKU demonstrates advantages in task comprehensiveness, cross-modal evaluation, and methodological generality:
>
> | Benchmark   | Modality       | Update Existing Knowledge | Unknown Knowledge Injection | Cross-Modal | General Capability | Evaluation Method   |
> | ----------- | -------------- | ------------------ | ----------------- | ----------- | ------------------ | ------------------- |
> | ZSRE        | Text           | ✗                  | ✗                 | ✗           | ✗                  | KE                  |
> | CounterFact | Text           | ✗                  | ✗                 | ✗           | ✗                  | KE                  |
> | MQuAKE      | Text           | ✗                  | ✗                 | ✗           | ✗                  | KE                  |
> | EvoWiki     | Text           | ✗                  | ✓                 | ✗           | ✗                  | KE                  |
> | VLKEB       | Multimodal     | ✗                  | ✗                 | ✗           | ✗                  | KE                  |
> | MMKE        | Multimodal     | ✗                  | ✗                 | ✗           | ✗                  | KE                  |
> | MMEVOKE     | Multimodal     | ✗                  | ✓                 | ✗           | ✓                  | RAG/SFT/IAG         |
> | **MMKU**    | **Multimodal** | **✓**              | **✓**             | **✓**       | **✓**              | **RAG/SFT/RLHF/KE** |
>
> We filter the data from three aspects: **knowledge validity, language quality, and image quality**,and only retain samples that are unanimously approved by three annotators (Unknown is reduced from 20K to 12.3K):
>
> To quantitatively verify reliability, we evaluate 50 sampled instances:
> - **47 samples** achieve complete agreement: including 32 unanimously accepted and 15 unanimously rejected.
> - **3 samples** show minor disagreement: all are cases where 2 annotators accepted and 1 rejected.
>
>
> Fleiss' Kappa is calculated as:
> $$\kappa = \frac{0.96 - 0.5648}{1 - 0.5648} \approx \mathbf{0.908}$$
>
> > **W2: Limited diversity of evaluated models.**
>
> We supplement evaluation results with representative models:
>
> |Model|Updated (Outdated)|Unknown (Correct)|
> |---|---|---|
> |ChatGPT-4o|72.41|7.20|
> |InternVL-2.5-78B|81.70|5.74|
> |Qwen2.5-VL-72B|84.90|7.27|
>
> - Larger models perform worse on **Outdated**: despite broader knowledge coverage, **knowledge competition** may weaken precise retention of specific outdated facts.
> - In **Unknown**, both large and small models show low accuracy (5.03%–7.27%), supporting the **temporal separation design** of the dataset.
>
>
>
> > **W3: Overstated theoretical contribution.**
>
> The phrase “providing theoretical guidance” may be imprecise. The main contribution of this work lies in **systematic experimental analysis**, and the explanations should be viewed as inductive interpretations of observed phenomena rather than a formal theoretical framework. We will revise the contribution statements and clarify the limitations.
>
> > **Q1: Can a static benchmark adequately capture the dynamic nature of real-world knowledge updates?**
>
> Regarding how a static benchmark captures the dynamic nature of knowledge, we clarify from the following perspectives:
>
> - MMKU does not aim to track real-time knowledge changes, but evaluates the capabilities of models and algorithms in a controlled setting, which have **long-term reference value**;
> - New updating strategies can be **repeatedly tested and compared** on the same static benchmark, enabling clear tracking of technological progress;
> - MMKU also establishes a **scalable data generation and validation pipeline**, maintaining its vitality through continuous updates and the introduction of new knowledge.
>
> > **Q2: How does existing knowledge in LLMs influence multi-modal knowledge updating performance?**
>
> We analyze the impact of embedded knowledge on multimodal updating from two aspects:
>
> - **Prior knowledge can reduce updating difficulty.** In the Unknown task, the model must simultaneously perform recognition, understanding, and internalization, often leading to incomplete answers; in contrast, existing knowledge helps quickly locate and absorb updated information.
>
> - **Existing knowledge may still cause interference.** Even with fine-tuning or editing, the model’s decision consistency in MCQ remains weaker than in Unknown (see Table below), indicating instability in parameterized updates.
>
> | Category | Model          | Full-FT | LoRA  | CPO   | ORPO  | SimPO | WISE  |
> | -------- | -------------- | ------- | ----- | ----- | ----- | ----- | ----- |
> | Evolving | InternVL2.5-8B | 34.70   | 48.16 | 34.47 | 40.49 | 35.04 | 34.70 |
> | Unknown  | InternVL2.5-8B | 45.12   | 47.42 | 38.99 | 47.09 | 48.28 | 38.65 |
> | Evolving | Qwen2.5-VL-7B  | 37.89   | 28.26 | 33.98 | 39.11 | 35.12 | 33.97 |
> | Unknown  | Qwen2.5-VL-7B  | 46.21   | 39.23 | 46.48 | 45.44 | 44.70 | 39.43 |

---

> > ### Author Rebuttal · Reviewer_ZZb7 · 2026-04-01
> >
> > I'm glad that my concerns have been addressed. I would like to keep my positive score, good luck!

---

> > > ### Author Response · Authors · 2026-04-07
> > >
> > > > **Dear Reviewer ZZb7,**
> > >
> > > Thank you very much for your thorough review and for your recognition of our work during the rebuttal stage. **Your suggestions are extremely valuable for improving the quality of our paper.**
> > >
> > > In response to the key issues you raised, we have made the following **substantial revisions** in the final version:
> > >
> > > 1. **Clear dataset comparison**: We have incorporated the dataset comparison table from the rebuttal into the revised manuscript to **help readers more quickly understand** the key improvements of MMKU.
> > >
> > > 2. **Strict quality control**: The **annotation protocol** of MMKU and the **Fleiss' Kappa** results have been added to the appendix. We also emphasize that only samples with complete agreement among all annotators are retained.
> > >
> > > 3. **Diverse evaluation models**: We have included three representative models, **ChatGPT-4o, InternVL-2.5-78B, and Qwen2.5-VL-72B,** as valuable reference points.
> > >
> > > 4. **Rigorous contribution statement**: We fully agree with your emphasis on academic rigor. The core value of this work lies in the systematic experimental analysis enabled by MMKU-Bench. We have **removed statements related to “theoretical guidance”** and consistently revised them to **“systematic experimental analysis.”** Additionally, we highlight the necessity of future theoretical research on knowledge updating in the future work.
> > >
> > > 5. **Value of a static benchmark**: We further clarify the contribution of MMKU as a static benchmark, which **supports the testing of new updating strategies and enables clear tracking of technological progress.** Moreover, the pipeline established by MMKU also supports continuous updates of benchmark knowledge.
> > >
> > > 6. **Impact of embedded knowledge**: Through Case Study and Decision Consistency analyses, we provide a deeper discussion on the dual impact of models’ prior knowledge on updating tasks **(facilitating internalization vs. causing decision interference),** thereby enhancing the depth of our analysis.
> > >
> > > We greatly appreciate each of your suggestions and are committed to completing the revision to the highest standard. We hope these concrete improvements address your concerns.
> > >
> > > **If you are satisfied with our revisions, we sincerely hope you would consider giving our work a more favorable final evaluation.** With your support, we believe MMKU can provide a more robust evaluation benchmark for the field of multimodal knowledge updating.
> > >
> > > Thank you again for your time and effort!
> > >
> > > Sincerely,
> > > **Authors of Paper 19771**

---

### Official Review · Reviewer_Mjyf · 2026-03-10

**Soundness:** 2
**Presentation:** 3
**Significance:** 2
**Originality:** 2
**Overall Recommendation:** 4
**Confidence:** 3

**Summary:**

This paper introduces MMKU-Bench, a benchmark tool for evaluating the knowledge updating ability of large multimodal models. Specifically, it distinguishes between updated knowledge (where the model must cover its existing outdated knowledge) and unknown knowledge (where the model learns previously unknown knowledge). The benchmark also evaluates cross-modal transfer and generality retention capabilities.

Experiments compare the performance of RAG, supervised fine-tuning, RLHF-based methods, and knowledge editing methods in these settings. Results show that updating existing knowledge is generally more difficult than learning unknown knowledge; improving knowledge injection capabilities often leads to a drop in general performance; and knowledge injected into one modality cannot always be reliably transferred to another.

**Compliance With Llm Reviewing Policy:**

Affirmed.

**Final Justification:**

I appreciate the authors’ additional experiments and the clear presentation of the results. These additions addressed my main concerns more convincingly, so I raise my overall recommendation from 3 to 4.

**Key Questions For Authors:**

Q1. How do the authors ensure that the comparison between updated and unknown knowledge is adequately controlled? The main argument of this paper relies on the differences between the two settings. However, based on the construction of the datasets, these two partitions differ not only in whether the model needs to update prior knowledge, but also in category distribution, entity type, question source, and construction process. This makes it difficult to determine whether the observed gaps are primarily caused by "updating existing knowledge" itself or by other distributional differences. Could the authors provide a more tightly controlled matching analysis that ensures consistency between updated and unknown samples in categories, templates, and answer formats, thus more clearly distinguishing the comparison results from expected factors?
Q2. To what extent does the updated portion reflect genuine knowledge updates, rather than adaptation to counterfactually edited samples? The updated portion in the benchmark is primarily constructed through counterfactual editing of existing examples, rather than stemming from naturally occurring real-world knowledge changes. This is understandable from a data collection perspective, but it raises concerns about ecological validity: performance on this split may partially reflect robustness to artificially edited examples rather than genuine knowledge updates. Could the authors clarify this limitation more clearly, and if possible, report results on a smaller subset of real-world updates from natural evolution to better validate the benchmark setting?

**Limitations:**

yes

**Strengths And Weaknesses:**

Strengths

S1.The problem setting is meaningful and practically relevant. This paper distinguishes between updated and unknown knowledge and explicitly studies the more challenging scenario of updating outdated knowledge stored in the model. This is a practical and realistic setting.
S2.The evaluation is relatively comprehensive, and the results are very insightful. In addition to knowledge injection performance, this paper also evaluates general capability retention and cross-modal transfer consistency. The results provide valuable insights into the difficulty of updating existing knowledge, the trade-offs with general performance, and the challenges of cross-modal knowledge transfer.

Weaknesses

W1. The core comparison between updated and unknown knowledge is not sufficiently controlled. The main conclusions of the paper rely on this comparison. However, the two splits appear to differ not only in whether prior knowledge must be overwritten, but also potentially in category distribution, entity types, and construction pipeline. As a result, the observed performance gap cannot be cleanly attributed to the difficulty of updating existing knowledge alone.
W2.The ecological validity of the updated split remains limited. The updated portion of the benchmark is mainly constructed through counterfactual editing, rather than being drawn from naturally evolving real-world knowledge. While this design choice is understandable for data collection, it raises the concern that the benchmark may partly measure adaptation to artificially edited examples rather than genuine knowledge updating ability.
3. 3.Some of the interpretations are stronger than the evidence directly supports. The paper provides fairly strong explanations in terms of old-knowledge interference, representational restructuring, and cross-modal misalignment. However, the current experiments mainly offer behavioral evidence, and the support for these mechanism-level interpretations is still limited.

---

> ### Author Rebuttal · Authors · 2026-03-31
>
> ## Dear Reviewer Mjyf,
>
> > **W1&Q1: The comparison between updated and unknown knowledge is poorly controlled, obscuring whether differences arise from updating difficulty or distribution bias.**
>
> We thank the reviewer for the insightful comment. To ensure a controlled and fair comparison between “updated knowledge” and “unknown knowledge,” we adopt the following designs:
>
> - **Unified construction pipeline:** Both types of data follow the pipeline in Section 3.2, maintaining consistency in QA generation and rewriting, CLIP-based semantic filtering, cross-modal construction, and quality evaluation, thereby reducing construction bias;
> - **Aligned statistical properties:** We strive to balance scale, subdomains, and length distributions to minimize the impact of statistical differences.
>
> However, considering that distribution differences may still exist in the initial collection stage, we further select subsets from the same categories for comparative analysis. The results are shown in the table below:
>
>
> | **Model**        | **Dataset** | **Stadium** | **Fish** | **Temple** | **Church** | **Plant** | **University** | Railway Station |
> | ---------------- | ----------- | ----------- | -------- | ---------- | ---------- | --------- | -------------- | --------------- |
> | **Internvl-2.5** | Updated     | 38.21       | 12.50    | 36.59      | 45.19      | 31.65     | 44.44          | 42.86           |
> |                  | Unknown     | 31.25       | 0.00     | 25.00      | 12.50      | 16.67     | 18.18          | 25.93           |
> | **Qwen2.5-VL**   | Updated     | 42.28       | 50.00    | 42.31      | 45.93      | 34.51     | 72.73          | 57.14           |
> |                  | Unknown     | 18.75       | 33.33    | 12.50      | 12.50      | 16.67     | 27.27          | 33.33           |
>
> Under the same subcategory conditions, the overall performance on the Updated task still exceeds that on the Unknown task. This indicates that the performance differences reported in the paper are not solely caused by category distribution or entity types, but are related to the nature of the knowledge itself.
>
>
>
> > **W2&Q2: Counterfactual edits in the updated split raise concerns about ecological validity and genuine knowledge updating.**
>
> - **Why adopt counterfactual editing?**
>     In the early stage of the study, we collected over 9K updated knowledge entries from Wikipedia, and only 4,223 remained after filtering (65.6% are person entities). After constructing VQA, the model accuracy was only 1.42%. Since the model could hardly answer correctly, it was difficult to verify whether it had learned the relevant knowledge. Moreover, the dataset was limited in size and unevenly distributed. Therefore, we adopted counterfactual editing to realize knowledge updating.
> - **Can counterfactual editing reflect knowledge updating?**
>     Counterfactual editing provides more controllable evaluation, with better scale and distribution, and avoids leakage from pretraining corpora. Meanwhile, constraints (same semantic category, non-synonymous, logically plausible) are applied to make it as close as possible to real updates.
> - **Validation on real-update subset:**
>     We additionally construct a small-scale real-update subset (60 samples) and conduct supplementary experiments. The results are shown below.
>
> | Updated        | Correct |  F1   | Outdated |
> | :------------- | :-----: | :---: | -------- |
> | InternVL2.5-8B |   --    |  --   | 100      |
> | SFT            |  61.67  | 60.78 | 20.00    |
> | LoRA           |  36.67  | 40.04 | 35.00    |
>
> | Updated       | Correct |  F1   | Outdated |
> | :------------ | :-----: | :---: | -------- |
> | Qwen2.5-VL-7B |   --    |  --   | 100      |
> | SFT           |  75.00  | 28.89 | 11.67    |
> | LoRA          |  56.67  | 23.77 | 25.00    |
>
> The models demonstrate stronger learning capability under real update scenarios, indicating that the performance differences do not stem from robustness to artificially edited data, but rather from the nature of different types of knowledge.
>
>
> > **W3: Experiments provide mainly behavioral evidence, with limited mechanistic support.**
>
> We agree that the current explanations regarding “interference from outdated knowledge,” “representation reconstruction,” and “cross-modal misalignment” are mainly based on behavioral evidence. We also clarify the positioning of this work: it focuses on **constructing a systematic evaluation benchmark** and, through large-scale experiments, **revealing key phenomena in multimodal knowledge updating**, rather than conducting rigorous causal mechanistic analysis.
>
> Therefore, we will explicitly state in the Limitations that these explanations are inductive observations. In addition, fine-grained mechanistic analysis will be an important future direction, and MMKU-Bench aims to provide a reliable evaluation foundation for such studies.

---

> > ### Author Rebuttal · Reviewer_Mjyf · 2026-04-03
> >
> > Thank you for the rebuttal. I have read the authors’ response and appreciate the clarifications.

---

> > > ### Author Response · Authors · 2026-04-03
> > >
> > > **Dear Reviewer Mjyf,**
> > >
> > > We sincerely thank you for carefully reading our rebuttal and for acknowledging that our responses have fully addressed your concerns.
> > >
> > > Your insights during the discussion phase have been invaluable, and have significantly motivated us to further refine and improve the manuscript.
> > >
> > > `We would like to highlight how your feedback has been incorporated into the final revision:`
> > >
> > > > **W1 & Q1: Distribution differences between updated and unknown knowledge.**
> > >
> > > To address the concern regarding potential distribution differences between “updated knowledge” and “unknown knowledge,” we have made the following revisions:
> > >
> > > - **Unified construction pipeline:**
> > >   In the revised manuscript (Section 3.3), we further emphasize that both types of knowledge follow a highly consistent construction process, including QA generation and rewriting, image semantic filtering, cross-modal knowledge construction, and quality evaluation. A unified standard is applied throughout to minimize biases introduced by the construction procedure.
> > >
> > > - **Alignment of scale and statistical properties:**
> > >   During data collection, we carefully balance the two sets in terms of dataset scale and statistical characteristics (e.g., subdomain distribution and sample length), thereby reducing potential confounding effects from distributional differences.
> > >
> > > - **Controlled subset validation:**
> > >   In addition, we have incorporated into the revised manuscript the category-matched subset analysis originally presented in the rebuttal. Under stricter control conditions, the results demonstrate that the observed performance gap primarily stems from the intrinsic nature of the knowledge itself, rather than distributional discrepancies.
> > >
> > > ---
> > >
> > > > **W2 & Q2: Ecological validity of counterfactual editing for knowledge updating.**
> > >
> > > We sincerely appreciate your understanding of our use of counterfactual editing in dataset construction. In response to this concern, we have made the following clarifications and additions:
> > >
> > > - **Motivation for counterfactual editing:**
> > >   At the early stage of our study, we attempted to collect real-world knowledge updates along with high-quality entity images. However, in the multimodal setting, since MLLM pretraining data is not accessible, when a model fails to answer a VQA instance correctly, it is difficult to determine whether the model has actually learned or updated the relevant knowledge. Therefore, to enable **controllable and verifiable evaluation**, we ultimately adopt counterfactual editing.
> > >
> > > - **Effectiveness of counterfactual editing:**
> > >   As clarified in our rebuttal, counterfactual editing provides a **more controlled and systematic perspective** for evaluating knowledge updating. Accordingly, we have expanded the discussion of this design choice in the revised manuscript to help readers better understand its strengths and limitations.
> > >
> > > - **Validation with real-world updates:**
> > >   Based on the real-world knowledge updates we collected, we conducted additional small-scale experiments to validate the main findings of the paper, and have included these results in the revised manuscript.
> > >
> > > - **Data availability:**
> > >   We are also willing to release this real-world update dataset to facilitate further research on multimodal knowledge updating.
> > >
> > > ---
> > >
> > > We believe that these revisions, directly inspired by your valuable feedback, have fully addressed the initial concerns and significantly improved the overall quality and rigor of the paper. **Given that the key issues have now been resolved, we sincerely hope you would consider whether the revised version warrants a more positive overall recommendation.**
> > >
> > > If you have any further questions, we would be happy to provide additional clarification.
> > >
> > > Sincerely,
> > > **Authors of Paper 19771**

---

### Official Review · Reviewer_egC6 · 2026-03-13

**Soundness:** 3
**Presentation:** 3
**Significance:** 2
**Originality:** 2
**Overall Recommendation:** 4
**Confidence:** 3

**Summary:**

The paper proposes MMKU-Bench, a multimodal knowledge updating benchmark covering 25k knowledge instances and over 49k images. The benchmark consists of (1) counterfactual constructed updating knowledge, (2) new unknown knowledge via comparing the difference between the Wikipedia November 2025 dump and the February 2025 dump. Evaluation includes SFT, RLHF, and KE methods and provides observations in knowledge injection, general capability, and consistency analysis.

**Compliance With Llm Reviewing Policy:**

Affirmed.

**Final Justification:**

The authors have addressed my concerns. Given my low confidence in the knowledge editing field, I still have concerns about counterfactual designs, but it is a common practice in the field, and the authors' response about "closer to real-world continuous update scenarios" addresses the ripple effect problem. My major concerns are addressed.

**Key Questions For Authors:**

See Weaknesses.

**Limitations:**

Yes.

**Strengths And Weaknesses:**

Strengths:
1. The knowledge updating differs from knowledge editing in that it first inspects the knowledge inside the MLLM by selecting the samples that can be correctly answered by MLLMs.
2. The benchmark and the evaluation are comprehensive, demonstrating a solid empirical foundation.

Weaknesses:
1. The counterfactual approach introduces severe logical conflicts. It is understandable that evaluating knowledge updating requires MLLMs to change their parametric knowledge, which may necessitate counterfactual editing since real-world knowledge cannot validate the effectiveness of the editing procedure. However, editing with counterfactual knowledge contradicts other related and unedited facts, triggering a "ripple effect". For instance, changing a bird's habitat from Canada to Germany without simultaneously updating its continent from North America to Europe creates conflicts. Consequently, the observed degradation in the MLLMs' general capabilities is expected since contradictory knowledge is involved. Therefore, this evaluation setting may not directly present the MLLM capability for self-consistent, reasonable, logically grounded knowledge updating in real-world scenarios.
2. Moreover, since most counterfactual knowledge is fake facts, the evaluation can be interpreted with bias. For example, whether (1) the MLLMs cannot support knowledge injection, or (2) MLLMs are robust to noisy and polluted knowledge, cannot be determined from this evaluation setting.
3. The fake knowledge in the benchmark necessitates the implementation of rigorous safety mechanisms to prevent data poisoning to the pretraining and SFT of real-world MLLMs.
4. Is the Unknown Knowledge also counterfactually edited? The real-world scenario is more like the Unknown Knowledge from the wikipedia update.
5. True knowledge updating should incorporate timestamps. As illustrated in the paper's own Figure 1, real-world knowledge often evolves and becomes time-bound rather than simply incorrect. Without explicit temporal grounding in the evaluation data, the difference between updating and editing is incremental, reducing the difference merely to the preliminary inspection of the MLLM's internal knowledge.

Even without this inspection step, the pretraining knowledge sources of the MLLM remain identical for both editing and updating tasks; thus, the inspection of whether the MLLM can answer the question correctly cannot accurately detect whether the knowledge is internalized in the MLLM; the wrongly answered question could be due to sampling or decoding strategies rather than has no knowledge.

6. The difference between MMKE and MMKU should be clarified. The "for diverse visual knowledge" is not well-illustrated in this paper.

---

> ### Author Rebuttal · Authors · 2026-03-31
>
> ## Dear Reviewer egC6,
>
> > **W1: The counterfactual approach introduces severe logical conflicts.**
>
> We further clarify the motivation and rationale behind this design:
>
> - Real multimodal update data is costly to obtain; therefore, we adopt **counterfactual editing** as a controllable and scalable alternative;
> - The ripple effect is a typical characteristic of real-world updates (e.g., chain changes caused by company renaming), used to examine the model’s conflict handling and consistency after local modifications;
> - MMKU-Bench does not pursue full self-consistency, but instead focuses on **robustness and stability** under incomplete updates, which is closer to real-world continuous update scenarios.
>
>
> > **W2: Counterfactuals obscure whether MLLMs fail at knowledge injection or simply exhibit noise robustness.**
>
> We understand this concern and clarify as follows:
>
> - Counterfactual knowledge is not random noise, but structured substitutions that satisfy **the same semantic category, non-synonymous expressions, and logical plausibility**, used to simulate real knowledge updates;
> - The significant differences between Updated and Unknown tasks in Table 2 indicate that the model does not simply treat them as noise;
> - Experiments further show that knowledge updates induce **structural changes** in model capabilities, which is more consistent with “knowledge reconstruction” rather than “noise suppression.”
>
>
> > **W3: Counterfactual knowledge raises safety concerns by enabling potential data poisoning.**
>
> We agree that indiscriminate use of counterfactual knowledge may introduce data poisoning risks. We clarify as follows:
>
> - MMKU-Bench is designed to evaluate the knowledge updating capability of multimodal models, rather than serving as training data for general pretraining or large-scale SFT.
> - To avoid potential misuse, we will explicitly clarify its scope and risks in the revised version and recommend against directly using counterfactual data for general model training, in order to reduce contamination risks.
>
> > **W4: Is the Unknown Knowledge also counterfactually edited?**
>
> Unknown Knowledge is not constructed via counterfactual editing, but is derived from real-world temporal evolution data. Therefore, the benchmark includes two types of knowledge changes:
> - **Updated Knowledge**: updates to existing knowledge.
> - **Unknown Knowledge**: newly emerging knowledge in the real world.
>
> This design aims to reflect two types of knowledge change processes in real scenarios and supports comparative analysis in experiments.
>
> > **W5: Lack of timestamps and inability to accurately detect whether MLLM has internalized knowledge.**
>
> **Regarding temporal attributes:**
> We acknowledge that knowledge has temporal properties, but this work does not focus on incremental modeling. Instead, it addresses a more fundamental question: when $K_{old} \rightarrow K_{update}$, can the model achieve coverage and eliminate reliance on outdated knowledge? This setting is orthogonal to explicit temporal modeling. The revised version will further clarify this and introduce temporal information as a future direction.
>
> **Regarding knowledge internalization:**
> We acknowledge that a single round of QA is insufficient to fully verify “internalization.” To mitigate this, we adopt the following strategies:
> 1. Only select samples that baseline models can answer correctly and consistently, ensuring that the entity information has been pre-internalized;
> 2. Explicitly evaluate the model’s reliance on outdated knowledge, revealing potential failure to update;
> 3. Design diverse evaluations (e.g., VQA and MCQ) to reduce biases introduced by a single decoding format.
>
> > **W6: The distinction between MMKE and MMKU is unclear, and the concept of “diverse visual knowledge” is insufficiently explained.**
>
> **The distinction between MMKE and MMKU:**
>
> - **Task:** MMKE focuses on local editing of a small number of facts; MMKU focuses on large-scale continuous knowledge updating and coverage.
> - **Setting:** MMKE does not distinguish whether knowledge has been internalized; MMKU is based on existing knowledge and distinguishes between updates and newly introduced knowledge.
> - **Evaluation:** MMKE evaluates editing accuracy; MMKU systematically compares RAG, SFT, RLHF, and KE methods, and introduces general capability and cross-modal consistency.
>
> **Diverse visual knowledge:**
> It is reflected in four aspects: multiple domains (331 subdomains), multiple types (updated+ unknown), cross-modality, and large scale (25k+ knowledge, 49k images).

---

> > ### Author Rebuttal · Reviewer_egC6 · 2026-04-03
> >
> > Thanks for the detailed response. My concerns are addressed. I will raise my overall recommendation from 3 to 4.

---

> > > ### Author Response · Authors · 2026-04-07
> > >
> > > > **Dear Reviewer egC6,**
> > >
> > > We are pleased that our response has addressed your concerns. We sincerely appreciate your careful review and valuable suggestions, and **we would like to further clarify and alleviate your concerns:**
> > >
> > > 1. **Why is it necessary to ensure that the model can answer correctly?**
> > >     Since the pretraining data of MLLMs is not accessible, when a model fails to correctly answer a VQA instance, we cannot determine whether it has actually learned the relevant knowledge.
> > >
> > > 2. **Why is it difficult to collect real knowledge updates?**
> > >     Constructing real data for evaluating knowledge updates is highly challenging, especially in multimodal scenarios, where it is difficult to simultaneously satisfy the following three conditions:
> > >     (1) A clear “old knowledge → new knowledge” mapping exists;
> > >     (2) There are visual entities that strictly correspond to the knowledge;
> > >     (3) The model has already stably internalized the knowledge (i.e., can correctly answer related questions).
> > >
> > > 3. **Why adopt counterfactual editing?**
> > >     In the early stage of our study, we collected over 9,000 updated knowledge entries from Wikipedia, and after filtering, only 4,223 remained (65.6% of which are personal entities). After constructing VQA data, the model achieved an accuracy of only 1.42%. Since the model was almost unable to answer correctly, and the dataset was limited in size and unevenly distributed, we decided to adopt counterfactual editing to **provide a more controlled and systematic perspective** for evaluating knowledge updates.
> > >
> > > 4. **Can counterfactual editing reflect real knowledge updates?**
> > >     Counterfactual editing offers stronger controllability, allowing coverage of larger-scale and more widely distributed data, while also avoiding leakage from pretraining corpora. In addition, we follow these principles when constructing answers:
> > >     (1) They belong to the same semantic category as the original answers;
> > >     (2) They are not synonymous or near-synonymous expressions;
> > >     (3) They remain linguistically and logically reasonable.
> > >     Therefore, this construction method stays **as close as possible to the characteristics of real-world knowledge updates**, while ensuring **controllability**.
> > >
> > > 5. **Regarding the ripple effect**
> > >     We agree that updating a single piece of knowledge can trigger a chain reaction, leading to inconsistencies with related knowledge, which is an inevitable phenomenon during knowledge injection into models. **Due to the diverse and complex relational structure of knowledge, it is difficult to exhaustively update all related knowledge simultaneously during post-training.** Therefore, the design of MMKU is closer to the continuous knowledge updating process in real-world scenarios. Meanwhile, the ripple effect is also an important research direction. As pointed out in [1], the issue of indirect probing failure in models is noteworthy, and we look forward to more work exploring this area.
> > >
> > >
> > > If you believe there are any aspects in which the paper could be further improved, we would be more than happy to continue the discussion. Thank you again for your time, support, and constructive feedback.
> > >
> > > Sincerely,
> > > **Authors of Paper 19771**
> > >
> > > **[1]** Li A O, Goyal T. Memorization vs. reasoning: Updating LLMs with new knowledge[C]//Findings of the Association for Computational Linguistics: ACL 2025. 2025: 25853–25874.

---

### Decision · Program_Chairs · 2026-04-30

**Decision:**

Accept (regular)

**Comment:**

This paper proposes the MMKU benchmark, constructing a multimodal knowledge update benchmark containing over 49K images. During the review process, the main discussions were as follows:

+ Reviewer egC6 expressed concerns regarding the counterfactual method, logical inconsistencies, and noise robustness. Furthermore, security issues and the handling of unknown knowledge also raised concerns.

+ Reviewer Mjyf primarily expressed that the comparison between updated and unknown knowledge remains limited. Moreover, the construction method is based on existing examples for counterfactual construction, lacking sufficient real-world knowledge changes. The reviewer acknowledged the authors' responses during the rebuttal process.

+ Reviewers ZZb7 and UAsc also raised concerns regarding quality assurance metrics, annotation quality, comparison with baseline methods, and the comprehensiveness of the evaluation method. All reviewers confirmed that these concerns were fully addressed during the response process.

The area chair summarized the above comments and requested the authors to rigorously revise the final version, ensuring the quality of the dataset, the evaluation method, and the details of the dataset construction. The area chair held a slightly positive opinion of the paper.